# Dynamic layer-specific processing in the prefrontal cortex during working memory

Jonas Karolis Degutis [1,2,3] ✉, Denis Chaimow [4], Daniel Haenelt [4], Moataz Assem[5], John Duncan [5], John-Dylan Haynes[1,2,3,6,7,8], Nikolaus Weiskopf [1,4,9,10] & Romy Lorenz[4,11] ✉

The dorsolateral prefrontal cortex (dlPFC) is reliably engaged in working memory (WM) and comprises different cytoarchitectonic layers, yet their functional role in human WM is unclear. Here, participants completed a delayed-match-to-sample task while undergoing functional magnetic resonance imaging (fMRI) at ultra-high resolution. We examine layer-specific activity to manipulations in WM load and motor response. Superficial layers exhibit a preferential response to WM load during the delay and retrieval periods of a WM task, indicating a lamina-specific activation of the frontoparietal network. Multivariate patterns encoding WM load in the superficial layer dynamically change across the three periods of the task. Last, superficial and deep layers are non-differentially involved in the motor response, challenging earlier findings of a preferential deep layer activation. Taken together, our results provide new insights into the functional laminar circuitry of the dlPFC during WM and support a dynamic account of dlPFC coding.

The prefrontal cortex (PFC) is critical for a diverse range of higher-level cognitive processes including working memory (WM)[1]. Early work on WM focused on the neural instantiation of stimulus-specific WM activity in the PFC of non-human primates[2,3], with later studies finding content-specific signals across multiple areas of the human cortex[4–7]. In a parallel line of work, instead of elucidating WM contents, studies examined neural activity related to WM load[8–11] (the number of remembered items) and WM manipulation (the reordering of stored information)[12]. These studies found a positive relationship between PFC activation and task demand for WM as well as other cognitive processes, suggesting that the PFC plays a critical role in the multiple-demand system, a frontoparietal network that commonly responds to a particular diverse range of different cognitively challenging tasks[13,14]. More recent studies have shown that multivariate activity patterns of the PFC dynamically change depending on WM demands[15], with these changes potentially being driven by altered coupling between the PFC and other key brain regions that process WM load[16,17].

Despite the PFC's importance to WM, the role of its laminar circuitry in relation to human cognition remains unclear. Evidence from non-human primates indicates that the PFC has cytoarchitectonic layers that play distinct roles in WM[18,19]. Superficial layers have been hypothesized to underlie the maintenance of WM[20], while deep layers are considered the output layers that send signals away from the PFC towards motor and premotor areas[21,22]. Recent advances in high spatial resolution functional MRI (fMRI) at ultra-high field strength have enabled the study of lamina-specific responses in human participants in a non-invasive manner with studies predominantly investigating sensory regions[23–28], and only recently the left dorsolateral PFC (dlPFC)[29]. In a double-dissociation, Finn et al. [29]. found that superficial layers of the dlPFC were preferentially activated by the manipulation of verbal WM compared to its maintenance, while deep layers responded to a motor response manipulation.

Here, we expand on these human laminar fMRI findings by addressing several open questions about the superficial and deep layers of frontal cortex, while focusing on the aforementioned left dlPFC. First, we manipulate WM load to see whether the superficial layer result from Finn et al. [29]. generalizes to other demanding high-level cognitive tasks. Second, we seek to replicate the preferential activation of deep layers to a motor response. Third, drawing

[1]Department of Psychology, Humboldt-Universität zu Berlin, Berlin, Germany. [2]Bernstein Center for Computational Neuroscience Berlin and Berlin Center for Advanced Neuroimaging, Charité Universitätsmedizin Berlin, corporate member of the Freie Universität Berlin, Humboldt-Universität zu Berlin, and Berlin Institute of Health, Berlin, Germany. [3]Max Planck School of Cognition, Leipzig, Germany. [4]Department of Neurophysics, Max Planck Institute for Human Cognitive and Brain Sciences, Leipzig, Germany. [5]MRC Cognition and Brain Sciences Unit, University of Cambridge, Cambridge, Cambridgeshire, UK. [6]Research Training Group "Extrospection" and Berlin School of Mind and Brain, Humboldt-Universität zu Berlin, Berlin, Germany. [7]Research Cluster of Excellence "Science of Intelligence", Technische Universität Berlin, Berlin, Germany. [8]Collaborative Research Center "Volition and Cognitive Control", Technische Universität Dresden, Dresden, Germany. [9]Felix Bloch Institute for Solid State Physics, Faculty of Physics and Earth Sciences, Leipzig University, Leipzig, Germany. [10]Wellcome Centre for Human Neuroimaging, Institute of Neurology, University College London, London, UK. [11]Max Planck Institute for Biological Cybernetics, Tübingen, Germany. ✉e-mail: j.karolis.degutis@maxplanckschools.de; romy.lorenz@tuebingen.mpg.de

from prior research on the impact of WM load on the strength and directionality of coupling between the dlPFC and other brain areas[16,17], we explore whether WM load induces layer-specific neural activity pattern alterations by investigating the multivariate code underlying WM load in superficial and deep layers. Fourth, we investigate the stability of this code across the entire duration of the trial, aiming to discern whether well-established dynamic coding properties of the PFC[30–34] localize to specific layers.

While dynamic coding more commonly refers to changes of the coding format of WM content[4,30,31,34], a broader definition of the concept has been used in the past to investigate dynamic changes in multivariate activity underlying different trial types[31] or to characterize different WM control processes at different trial stages[15]. Here, we follow this broader notion of dynamic coding and employ multivariate decoding of WM load to investigate WM control processes across three periods of the WM task (encoding, delay, retrieval) with the aim to understand whether the multivariate patterns underlying these control processes change in a layer-specific manner.

To preview, we found that WM load preferentially activated superficial layers compared to deep layers during the delay period, and we observed higher load decoding accuracy in superficial layers during the retrieval period. We did not find a significant difference between deep and superficial layers to the motor response manipulation. Finally, we observed that multivariate patterns underlying load at different periods of the WM task changed across time in the superficial layers of the left dlPFC.

## Results

Nine participants were scanned at 7 T field strength using a GE-BOLD sequence and used their right hand to complete a WM delayed match-to-sample task, where in two of the four runs we manipulated WM load. Participants had to remember either four or one item and had to always respond during the retrieval period (Fig. 1a). In the other half of the runs the motor response was manipulated: participants had to always remember four items and during the retrieval period they were asked to respond or abstain from responding to a presented probe (Fig. 1a). Participants accurately remembered the stimulus in both the low load (mean = 95.5% correct, standard deviation (SD) = 6.34%) and high load condition (mean = 74.3% correct, SD = 10.6%; Supplementary Fig. 1, left panel) with the high load trials having significantly

lower accuracy ($p < 0.001$, $t(8) = 9.44$, $CI_{95} = [16.8, 27.7]$, $d = 3.15$, two-tailed paired t-test). During the motor manipulation, the participants followed instructions by responding (mean = 99.3% correct, SD = 2.08%) and abstaining to respond (mean = 97.9% correct, SD = 3.13%; Supplementary Fig. 1, right panel) with no significant difference between motor conditions ($p = 0.35$, $t(8) = 1.00$, $CI_{95} = [-1.81, 4.59]$, $d = 0.33$, two-tailed paired t-test).

To allow a systematic comparison with the findings reported by Finn et al.[29], all our analyses focused on the left dlPFC (Fig. 1b). To assess the specificity of our results, we additionally looked at a set of anatomically adjacent frontal regions in the left hemisphere from the cingulo-opercular network as a control (Fig. 1b). We also report results from the right dlPFC in the Supplementary Material due to its involvement in WM processing and as it was covered in our fMRI acquisition slab. As a control for the right dlPFC, we also report results from right-lateralized frontal regions from the cingulo-opercular network. Please note though, that we a priori decided to specifically investigate the left dlPFC and as such optimized our signal in the left hemisphere (see *Methods*). We found no significant difference in the number of voxels per layer within a given ROI (Supplementary Fig. 4), allowing comparability of our results across different layers and regions.

All results in the main text are presented for trial periods that are adjusted for a hemodynamic delay of 6 s while "true" trial timings (without taking into account the hemodynamic delay) are additionally indicated in Figs. 2 and 3.

### Superficial layers preferentially activate to higher WM load during the delay period

Our first goal was to examine the influence of WM load on the superficial and deep layer activity of the left dlPFC (Fig. 1b, c). We predicted that superficial layers would be preferentially influenced by WM load. Trial time courses were estimated for each load condition in each layer in the dlPFC and frontal control regions (Fig. 2a, c). To examine the load effect, we subtracted the low load time course from the high load for a given layer in a given region and examined two trial periods of interest: middle of the delay period (11.3–15.1 s), which was after the encoding peak and before the response probe, and retrieval (20.7–24.5 s; indicated as two gray shaded rectangles in Fig. 2a, c), which was after the presentation of the probe.

**Fig. 1 | Trial design and ROIs. a** Trial design. In load runs (left), participants performed a delayed match-to-sample task, where they had to remember a presented high or low load stimulus (four vs. one item) and had to indicate whether the probe presented at the end of the trial was a part of the stimulus array or not. In motor runs (right), participants were always presented with four items. If an 'X' was presented during the probe, they had to abstain from answering. High load stimulus depicts four outdoor scenes. Low load stimulus depicts one outdoor scene and three masks. **b** Regions of interest. The left dlPFC was defined as four parcels from the frontoparietal network: 8 C, IFJp, IFSa, p9-46v[76,83]. The left control regions were defined as four parcels from the cingulo-opercular network anatomically adjacent to the left dlPFC: FEF, 6r, 46, FOP5[76,83] (for full description of ROIs see Methods). **c** Three equidistant gray matter layers (only superficial and deep layers were used for analyses) defined in the space of the functional images were projected onto an anatomical T1 and functional T2* image of an example participant.

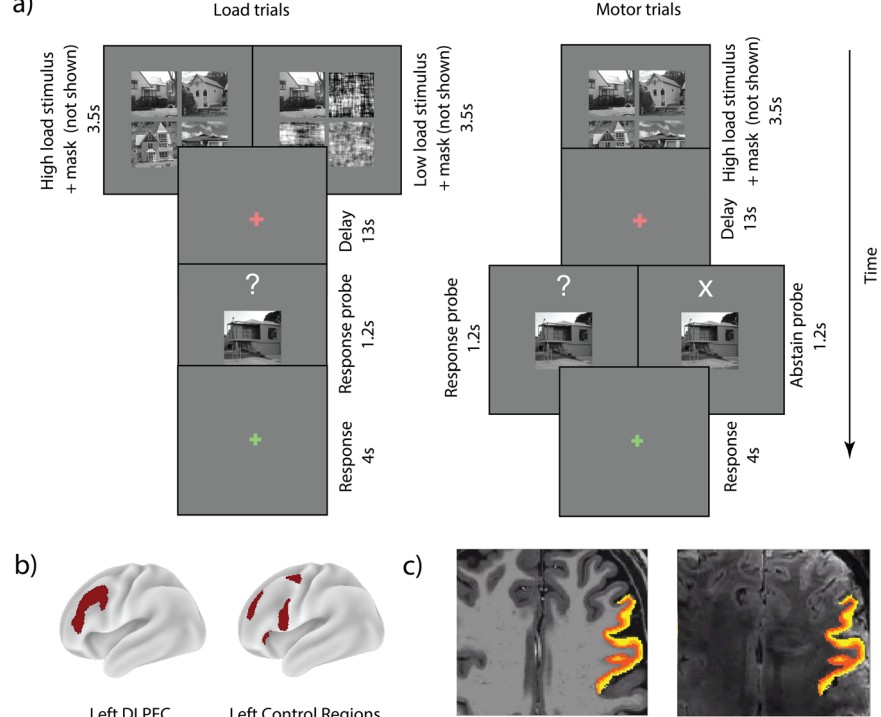

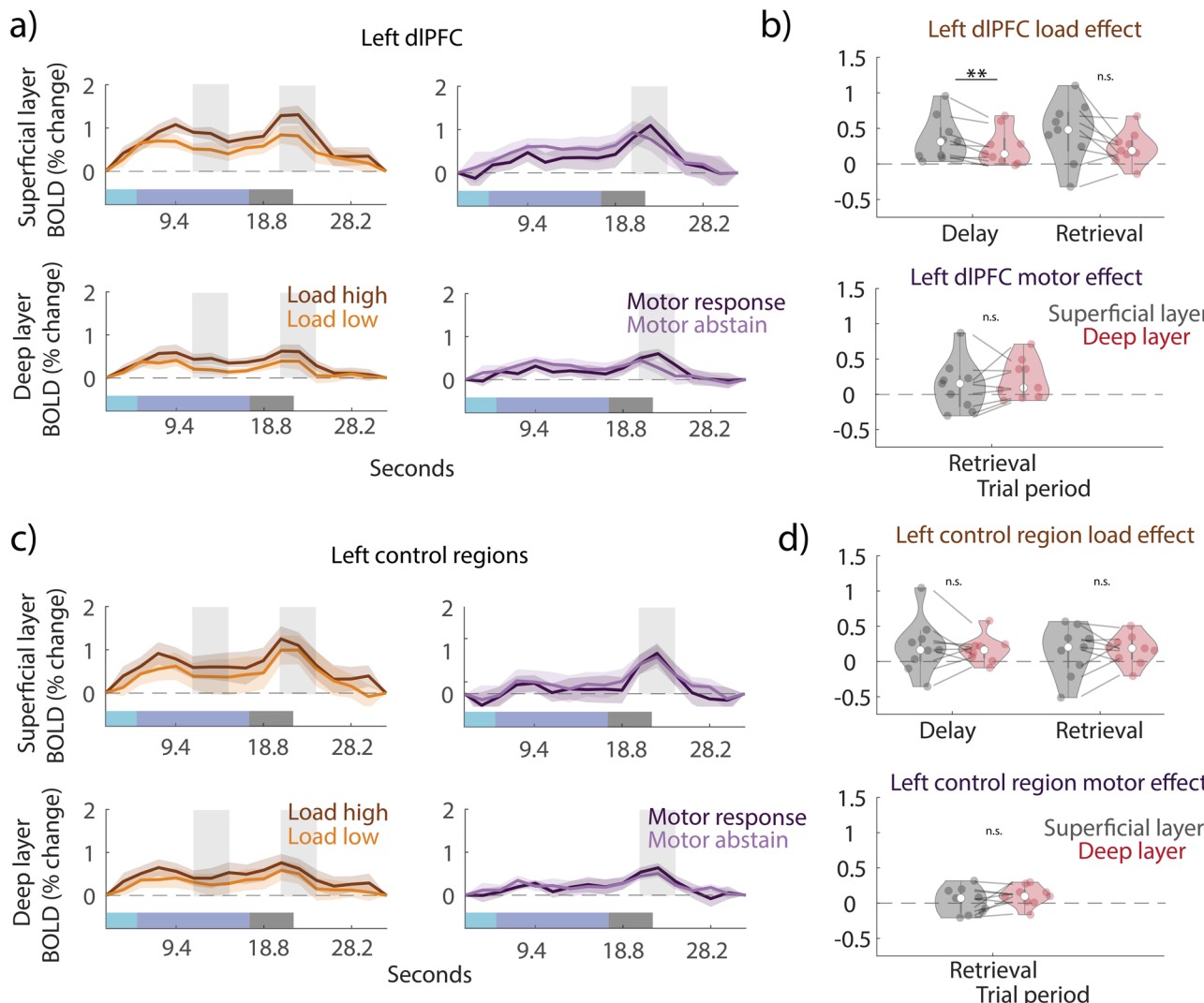

**Fig. 2 | Layer-specific univariate BOLD responses. a** Univariate trial time courses averaged across all voxels from the superficial (top) and deep (bottom) layers in the left dlPFC for load trials (high and low conditions; left plots) and motor trials (response and abstain conditions; right plots). The vertical gray transparent rectangles indicate the two trial periods of interest (adjusted for the hemodynamic delay of 6 s): the delay (11.3–15.1 s) and the retrieval period (20.7–24.5 s). The shaded area indicates standard errors across participants. The horizontal colored transparent rectangles at the bottom of the figure indicate "true" trial periods (without taking hemodynamic delay into account) in the following order: stimulus presentation (light blue), delay period (blue), probe presentation and retrieval period (gray). **b** The layer-specific load effect (top) and motor effect (bottom) in the superficial and deep layers of the left dlPFC for the delay and retrieval time periods. The load effect was calculated per subject by subtracting the low from the high load time course and averaging time points associated with the trial periods of interest. The motor effect was calculated per subject by subtracting the abstain from the motor response time course and averaging time points in the retrieval period. The stars indicate $p < 0.01$ (two-tailed paired t-test). Error bars indicate ±SEM. **c** same as (**a**) but for left control regions. **d** Same as (**b**) but for left control regions.

We found a higher load effect across participants in superficial as compared to deep layers during the delay period within the left dlPFC ($p = 0.0072$, $t(8) = 3.58$, $CI^{95} = [0.040, 0.188]$, $d = 1.19$, two-tailed paired t-test), but no significant difference during the response ($p = 0.072$, $t(8) = 2.07$, $CI^{95} = [-0.024, 0.46]$, $d = 0.69$) (Fig. 2b). In left control regions we found no difference between layers for the load effect in either the delay ($p = 0.55$, $t(8) = 0.630$, $CI^{95} = [-0.14, 0.24]$, $d = 0.21$) or the retrieval period ($p = 0.83$, $t(8) = -0.223$, $CI^{95} = [-0.20, 0.16]$, $d = -0.07$) (Fig. 2d). Preferential superficial layer activation to high WM load during the delay and retrieval periods was significant in the right dlPFC (Supplementary Fig. 2b) and right control regions (Supplementary Fig. 2d).

### No differential layer activation to motor response in retrieval period

Additionally, we examined the motor effect during the retrieval period (20.7–24.5 s). We expected to replicate higher activation of the deep layers[29]. Similarly to the load effect, we subtracted the abstain from the motor

response time course for each layer in each ROI (Fig. 2a, c). However, we found no significant difference between superficial and deep layers either in the dlPFC ($p = 0.30$, $t(8) = -1.11$, $CI^{95} = [-0.268, 0.094]$, $d = -0.37$) (Fig. 2b) or in the control regions ($p = 0.18$, $t(8) = -1.452$, $CI^{95} = [-0.20, 0.045]$, $d = -0.48$) (Fig. 2d). We also found no significant motor effect difference between layers in the right dlPFC or right control regions (Supplementary Fig. 2b, d).

### Superficial layers preferentially code for WM load in retrieval period

Univariate analyses demonstrate changes in average signal between layers; however, individual voxel responses within each layer are not necessarily homogeneous. To be more sensitive to such patterns we used multivariate decoding analyses (for examples in GE-BOLD see refs. 25,35,36) using trial normalization (see *Methods*) that mitigated the effect of differences in mean univariate response between the conditions which might drive the decoding results. This analysis allowed us to investigate multivariate activation

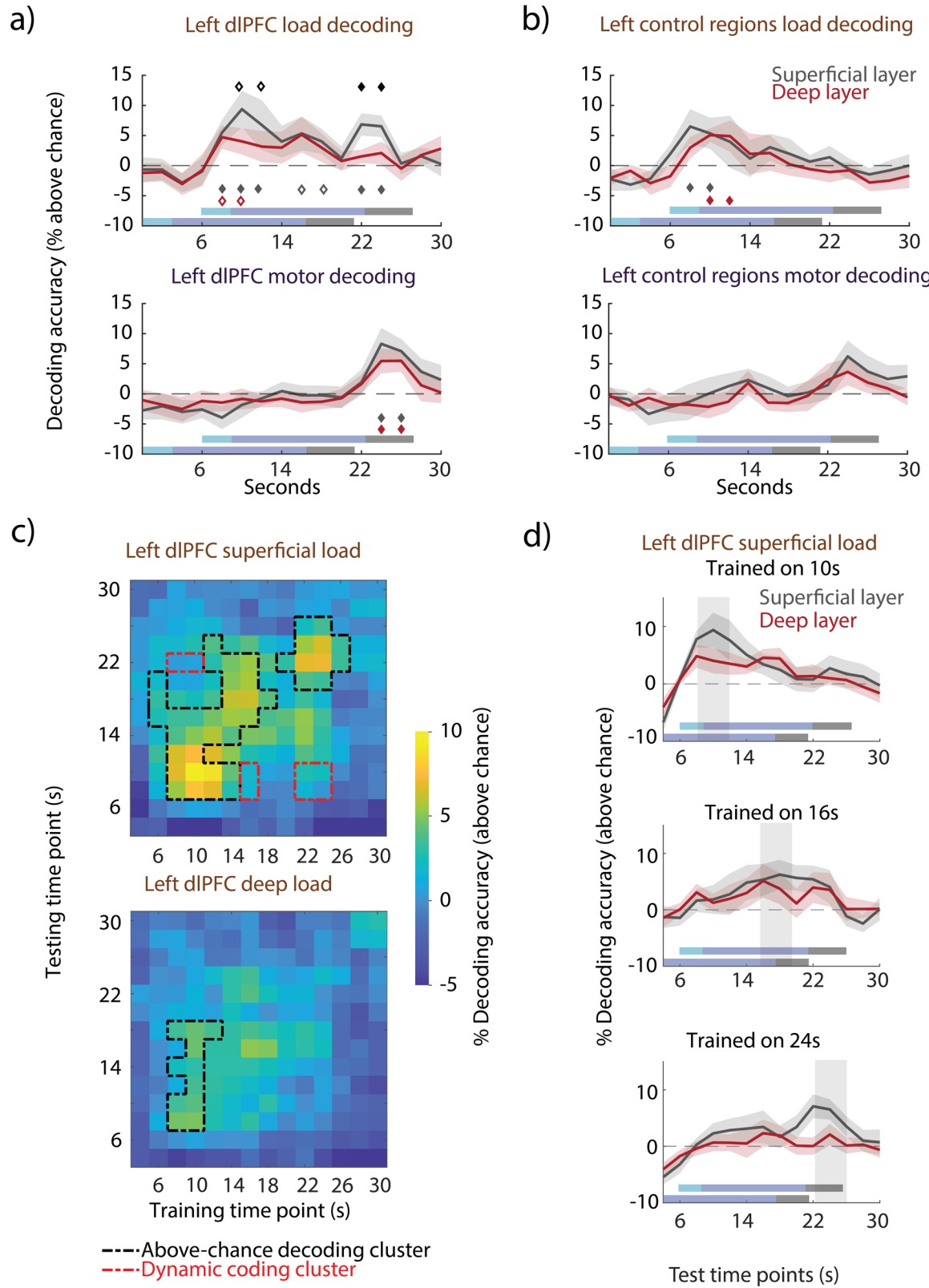

patterns of different layers to load and motor conditions as well as potential differences in response patterns over the course of the trial. Therefore, unlike in the univariate analysis where we compared responses averaged over delay and retrieval periods of the task, here we ran the classification in an exploratory manner across the length of the trial, encompassing the encoding, delay, and retrieval periods.

Previous studies have used multivariate decoding to investigate control processes of WM memory using load[15,31,37–40]. Here we followed these studies and trained a binary classifier to differentiate between high and low load trials independently for each layer within each ROI and statistically compared decoding results against permutation-derived null distributions as well as between layers. For superficial layers of the left dlPFC, we found two

**Fig. 3 | Layer-specific decoding accuracy. a** Top: decoding accuracy (linear SVM) of high vs. low load conditions across time from the superficial (gray) and deep layers (red) of the left dlPFC. Bottom: decoding accuracy of response vs. abstain motor conditions across time from the superficial (gray) and deep (red) layers of the dlPFC. Diamonds indicate above-chance decoding significance of superficial and deep layers (in their corresponding color) and comparison between layers (black). Filled diamonds depict significance of $p < 0.05$, (cluster-permutation test). Hollow diamonds depict $p < 0.1$, (cluster-permutation test). The shaded rectangles indicate the trial periods in the following order: stimulus presentation (light blue), delay period (blue), probe presentation and retrieval period (gray). "True" trial periods (without accounting for hemodynamic delay) are displayed at the bottom and trial periods adjusted for a 6 s hemodynamic delay are displayed on top. **b** same as (**a**) but for left control regions. **c** Temporal cross-decoding of load from the superficial and deep layers of the dlPFC. Top: superficial layers. Bottom: deep layers. A classifier trained on each time point was tested on itself and all other time points. The matrix diagonal corresponds to the classification accuracy of load in (**a**). The black outline denotes all above-chance clusters within a layer (nonparametric cluster-permutation test against null, $p < 0.05$). The red outline signifies all dynamic clusters (conjunction of two nonparametric cluster-permutation tests, $p < 0.05$; see *Methods: Temporal cross-decoding*). **d** Selected columns from (**c**) visualized in 2D as decoding time courses: accuracy across the trial when trained on the encoding (top plot; 10 s), delay (middle plot; 16 s), and retrieval (bottom plot; 24 s) trial periods and tested on all time points in turn. Columns at 10 s, 16 s, and 24 s in (**c**) correspond to the top, middle, and bottom time courses, respectively. Transparent gray vertical bars indicate the time points that the classifier was trained on. Shaded area depicts ± SEM. To increase signal-to-noise, all decoding points were averaged with their preceding one in a moving average manner.

above-chance temporal decoding clusters during the encoding and early delay period (8-12 s, $p = 0.0035$, $d_{cluster\ \bar{X}} = 0.86$, one-tailed permutation test) as well as at retrieval (22-24 s, $p = 0.0027$, $d_{cluster\ \bar{X}} = 1.34$, one-tailed permutation test) (Fig. 3a, filled gray diamonds) while two above-chance time points during the late delay failed to reach significance on the temporal cluster-level (16-18 s, $p = 0.077$, $d_{cluster\ \bar{X}} = 0.67$, one-tailed permutation test) (Fig. 3a, hollow gray diamonds). For deep layers of the dlPFC, two above-chance time points during the encoding and early delay period trended towards temporal cluster significance (8-10 s, $p = 0.058$, $d_{cluster\ \bar{X}} = 0.75$, one-tailed permutation test; Fig. 3a, hollow red diamonds). For between layer comparison, we found two time points during the encoding and early delay period (10-12 s) where the superficial layer showed stronger WM load decoding than the deep layer, with this time period trending towards significance at a temporal cluster level ($p = 0.059$, $d_{cluster\ \bar{X}} = 0.85$, two-tailed permutation test; Fig. 3a, black hollow diamonds). Interestingly, we found one cluster that showed a significantly higher decoding of WM load in the superficial compared to the deep layer during the retrieval period (22-24 s, $p = 0.023$, $d_{cluster\ \bar{X}} = 1.08$, two-tailed permutation test; Fig. 3a, black filled diamonds) even though the bottom-up sensory input at retrieval was completely identical between the two conditions and consisted of a single probed image (Fig. 1a, response probe). These results were not driven by differences in accuracy (Supplementary Fig. 3a) and reaction time (Supplementary Fig. 3b) between low and high load trials. Moreover, we found a significant interaction between the three task periods of encoding (8–12 s), delay (14–20 s), and retrieval (22–24 s) and superficial and deep layers of the left dlPFC ($p = 0.034$, $\eta^2 = 0.35$, non-parametric repeated-measures ANOVA).

For the left control regions we found above-chance decoding clusters in the superficial (8–10 s, $p = 0.046$, $d_{cluster\ \bar{X}} = 0.77$, one-tailed permutation test) and deep layers (10-12 s, $p = 0.0043$, $d_{cluster\ \bar{X}} = 0.95$, one-tailed permutation test) during the encoding and early delay period (Fig. 3b), but no differences between layers nor an interaction between the three trial periods and layers ($p = 0.70$, $\eta^2 = 0.04$, non-parametric repeated-measures ANOVA). We also decoded WM load from the encoding and early delay period in the superficial layers and deep layers of the right dlPFC and found a difference between layers with higher superficial layer decoding, but no interaction between the three trials periods ($p = 0.97$, $\eta^2 = 0.005$, non-parametric repeated-measures ANOVA) (Supplementary Fig. 2e). We also decoded load from the right control regions during the encoding and early delay period but found no interaction between the three trial periods and layers ($p = 0.083$, $\eta^2 = 0.27$, non-parametric repeated-measures ANOVA) (Supplementary Fig. 2f).

### Both superficial and deep layers non-preferentially code for motor response in retrieval period

To detect patterns associated with the motor effect, we trained a binary classifier to distinguish between response and abstain trials. Since the participants only found out about the type of trial during the presentation of the probe, as expected, there was no significant decoding prior, either in the left dlPFC or left control regions. We found a single above-chance cluster each

in the superficial ($p = 0.0054$, $d_{cluster\ \bar{X}} = 1.27$, one-tailed permutation test) and deep layers ($p = 0.015$, $d_{cluster\ \bar{X}} = 1.18$, one-tailed permutation test) during the retrieval period (24-26 s; Fig. 3a, gray and red diamonds). However, there was no significant difference between the two layers. These results agree with our univariate findings showing no difference between superficial and deep layers during the motor retrieval period. The right dlPFC and right control region showed similar motor decoding results. Motor trial decoding was significant in both layers of the right dlPFC and right control regions during the retrieval period (Supplementary Fig. 2f).

### Dynamic coding of WM load in superficial layers

We sought to determine whether the multivariate pattern sustaining load information is stable across time. To do so, we used temporal cross-decoding, as previously done to investigate the stability of WM content in humans and non-human primates[30,31,41,42]. We trained on a given time point and tested on all time points in turn. This analysis provided us with two distinct types of elements within the cross-decoding matrix: the diagonal, which indicates the amount of information coding for load within a given time point, and the off-diagonal, which provides the generalization between two time points. If the trained classifier generalizes to other time points (i.e., high decoding accuracies of the off-diagonal elements of the decoding matrix), the analysis reveals that the multivariate pattern sustaining load information (i.e., "coding") is stable across time. However, when the diagonal of the matrix, representing training and testing on the same time points, shows higher decoding accuracies than the off-diagonal elements, then we can conclude from this lack of cross-generalization of the decoder, that the multivariate activity is dynamic and varies across time. We examined the generalization of multivariate patterns in response to load in the superficial and deep layers of the dlPFC.

Within the superficial layers of the dlPFC, we identified two above-chance decoding clusters (Fig. 3c, black dashed line). The first decoding cluster spanned both the encoding and delay period ($p = 0.001$, $d_{cluster\ \bar{X}} = 1.04$, one-tailed permutation test), while the second cluster was isolated to the retrieval period (23–25 s; $p = 0.003$, $d_{cluster\ \bar{X}} = 1.15$, one-tailed permutation test). Looking more closely into the first decoding cluster, we observed that the classifier trained on the encoding period (8–10 s) generalized to the middle of the delay period (16 s), and the classifier trained on the delay period (14–18 s) generalized to the late delay period (up to 22 s). Importantly though, the classifier trained on the encoding period did not generalize to the late delay period and vice versa, indicating a differential multivariate pattern sustaining load information in those two periods of the trial. To visualize these results, we plotted the columns from Fig. 3c as time courses (Fig. 3d, top panel), which depict that the classifier trained on 10 s had high decoding during the encoding and early delay, but the decoding accuracy waned in the second half of the trial. Similarly, the classifier trained on 16 s had the highest decoding during the late delay period (Fig. 3d, middle panel). With respect to the second decoding cluster identified at retrieval, we observe that it only generalizes to its surrounding time points, indicating that the multivariate pattern reflecting load during the retrieval period was distinct from encoding and delay (bottom time course in Fig. 3d).

Within the deep layers of the dlPFC, a single above-chance cluster (Fig. 3c, black dashed line, $p = 0.002$, $d_{cluster\ \bar{X}} = 1.10$, one-tailed permutation test) was found during the encoding period (8-10 s) which generalized to the delay (18 s). Plotting the column of Fig. 3c as a time course (Fig. 3d, top panel), we see that the deep layer results follow a similar trend as the superficial layer results: the classifier trained on 10 s had high decoding during the encoding and early delay, but the decoding accuracy waned in the second half of the trial. Similarly, the classifier trained on 16 s had the highest decoding during the late delay period (Fig. 3d, middle panel).

To empirically check the lack of generalization between the above-chance clusters, we identified dynamic elements within the cross-decoding matrices. Dynamic elements were calculated as a conjunction of above-chance elements surviving two cluster permutation tests (see *Methods* for details). Three clusters were found in the superficial layers of the left dlPFC: two off-diagonal elements at the intersection of the encoding and early delay (train) and retrieval (test) periods, two elements at the intersection of the middle delay (train) and encoding and early delay (test) periods, and four elements at the intersection of retrieval (train) and encoding and early delay (test) periods (Fig. 3c).

To summarize, our cross-decoding results showed three differential multivariate response patterns sustaining load in the superficial layers during the encoding, delay, and retrieval periods, indicative of dynamic coding between trial periods.

## Discussion

We scanned participants at ultra-high field strength performing a delayed match-to-sample task where we manipulated WM load and motor response while measuring responses from the superficial and deep layers of the dlPFC. In the left dlPFC, we found a higher superficial compared to deep layer activation in response to increased working memory (WM) load during the delay period; however, we did not find a differential laminar response to a motor manipulation. To examine the multivariate response across the length of a trial, we decoded load and motor trial types from the layers of the left dlPFC. We found that superficial layers preferentially code for difference in WM load during the retrieval period. With respect to the motor trials, we found that both superficial and deep layers code for the motor response in the retrieval period but failed to find any significant laminar difference. Finally, we also found that the multivariate code underlying WM load in the superficial layers dynamically changed between the encoding, delay, and retrieval periods.

Our results replicate early non-human primate studies showing persistent spiking in the lateral PFC in all periods of a WM task[2] by showing sustained univariate activity in both superficial and deep layers during high load trials. Our work also corroborates findings showing higher activation of the lateral PFC to higher task demands, as previously seen in univariate human fMRI work on WM load[8,9] and the manipulation of WM information[12]. Additionally, our results expand on previous WM literature that has used multivariate decoding to investigate neural mechanisms of WM[5,6,41-44].

We also provide insights into the mesoscale functional architecture of the dlPFC in response to WM load. Greater neural processing associated with WM load in superficial layers compared to deep throughout the delay and retrieval periods might indicate activity related to more general task demands; these demands increase as a result of higher WM load, possibly by both recruiting new neural populations (e.g., in retrieval) and by increasing univariate activity (e.g., in delay). These WM control processes might correspond to the updating of content during encoding, the monitoring or contents throughout the delay period[45], or the compression or reorganization of WM contents as a result of increased load demand[46]. Interestingly, there was a preferential univariate response of the superficial compared to deep layers to high load during the delay period, but no laminar difference in decoding. Superficial layers might utilize the same neural population underlying one or several control processes to a higher degree. This would result in higher univariate activity, but lower decodability since univariate contributions were minimized through trial normalization. Taking into account results from Finn et al.[29]., preferential response in the superficial layers to both WM manipulation and higher WM load might point towards a lamina-specific (i.e., superficial layer) activation of the frontoparietal multiple demand system[13] for heightened task demands more generally. However, it is essential to underscore the necessity for future investigations to systematically examine the layer-specific activation of the frontoparietal network across a diverse spectrum of cognitive tasks with varying demands; as can be accomplished employing novel real-time fMRI approaches utilizing neuroadaptive Bayesian optimization[47-49].

Previous studies have established that recurrent connections within superficial layers of the PFC support sensory input-specific gamma-band activity, while deep layer alpha and beta oscillations modulate this superficial layer activity[18,33]. In turn, the interaction between deep and superficial layers might act as a gating mechanism for sensory information to enter the PFC and WM storage[50,51]. In our study, both superficial layers and deep layers had above-chance temporal cross-decoding clusters during the encoding of WM information. Deep layer coding for WM load during the encoding period might correspond to the opening of the gate, which allows for the encoding of new WM information within the superficial layers. Based on our results, we can only speculate whether deep layers drive superficial layer activity. It should be noted that deep layer coding was significant in the right dlPFC and only trended towards significance in our temporal cluster analysis of the left dlPFC. This might indicate a lateralization in the WM encoding control process to the right hemisphere[52]; however, further studies are needed to corroborate these results and unravel any causal laminar relationships in both hemispheres.

For the retrieval period, we found significantly higher superficial compared to deep layer decoding of WM load which might indicate increased demands during the probe's comparison to the stored memory (1 vs. 4 items) despite the presentation of identical response probes in low and high WM load conditions. The comparison process seems to happen within the superficial layers, which might multiplex both the novel sensory probe stimulus and the remembered memoranda; to achieve this at high load, additional neural populations within the superficial layers might be recruited, thus driving the multivariate decoding. Superficial layer decoding relates to previous findings showing ramping gamma-band activity at the end of memory delays[53], as a consequence of its disinhibition resulting from the drop of alpha/beta power to allow a successful read-out of information from WM[54]. Intriguingly, load decoding peaked prior to motor decoding, which might denote the sequence of computations in the dlPFC: a comparison of the probe to the WM set before the initialization of a motor response.

Our results show multivariate laminar activity patterns related to load that are specific to the dlPFC. For the left dlPFC, we find that superficial layers show a significant decoding during the encoding and retrieval period and trend towards significance during the delay period. We find a higher superficial compared to deep layer decoding in the response period as well as trending towards significance in the encoding period. When comparing this to the left control regions, we only find above-chance decoding during the encoding period, but no effects for other trial periods. More importantly, no layer-specific results were identified in the left control region. We also find a significant interaction effect between the three trial periods and layers in the left dlPFC. The results from the right dlPFC show a preferential involvement of superficial layers during the encoding period. Despite the similar result seen in the right control regions, the laminar difference emerges before the onset of the encoding period and seems to be driven by the single negative time point in deep layers, which likely negates its significance.

Our findings of WM load-induced layer-specific changes in neural activity patterns in the left dlPFC can be contextualized by previous research in non-human primates and human fMRI studies. More specifically, it has been shown that increased WM load changes the strength and directionality of neuronal coupling between the prefrontal cortex, the frontal eye fields, and lateral intraparietal cortex[16], and influences both between-network and within-network coupling among frontoparietal, ventral attention, and default mode networks[17], which are reflected in a change in multivariate

pattern. This altered network coupling in the left dlPFC may result in subtle variations in neural activity patterns when WM demands change, which has been observed before[15] and which we localize to the superficial layers of the dlPFC. This observation suggests a network-coding perspective of WM and aligns with the established intra-prefrontal and cortico-cortical connections of the superficial layers in the prefrontal cortex[18,33]. In addition, changes in the multivariate patterns of the superficial layers in response to WM load may also be attributed to activation spread beyond initially activated regions, potentially recruiting new neural populations, although such signal spread might also manifest in univariate activation differences. Equally, the multivariate pattern in the dlPFC could suggest content-independent pointers[38] that project to parietal and sensory cortices where the maintained memoranda are thought to be stored[7,55,56].

The canonical model of the WM suggests that that information coded within the PFC is stable and firing is persistent across the delay period[3,57]. However, evidence has emerged that WM is both rhythmical and non-persistent in activation and the format of the coded information dynamically varies across time[33,58]. Studies using temporal cross-decoding have observed a dynamic code between the encoding period and delay activity in the PFC of non-human primates and humans[30,31,34], and across the visual hierarchy in humans[4,42]. We find three non-generalizing sub-clusters within the span of the trial; each sub-cluster corresponding to either the encoding, delay, or retrieval periods. Previous studies argued that dynamic coding indicates a point where transient sensory stimuli are transformed into a more stable representation[30]; however, since we were decoding load and not the contents of WM, the decoding sub-clusters might underlie separate WM control processes that occur on the WM content within the same layer.

In contrast to superficial layers of the PFC, the deep layers are considered to be the output - or top-down - layers of the cortical area and have neurons modulated by D2 dopamine receptors[21,22]. Finn et al.[29]. found that a motor modulation only activated the deep layers of the prefrontal cortex, while superficial layers stayed at baseline. We do not find a preferential activation of either superficial or deep layers of the left dlPFC to the motor manipulation either in univariate activity or in multivariate decoding. The discrepancy between Finn et al.[29]. and our results may be related to a difference in paradigm, a difference in scanning sequences, or a smaller sample size.

Finn et al.[29]. contrasted trials in which participants either had to respond after the presentation of a probe or had to not respond when a probe was not presented. This resulted in a change of two factors between conditions: the presence or absence of the probe stimulus and the motor response. Our paradigm always included the presentation of the probe and participants were asked to either respond or abstain from responding based on a presented cue. If the deep layer modulates the information that enters the superficial layer of the dlPFC, then that computation could provide an additional deep layer response. Thus, higher activation of the deep layers in Finn et al.[29]. might not be a result of the output, but rather of the opening of the gate by the deep layers to let in the probe for subsequent comparison in the superficial layers. Hence, this probe input signal would have been absent when contrasting between conditions in our paradigm, since participants could not predict whether a given trial required a response or not, they would have maintained the memoranda and automatically compared it to the presented probe. Based on our results we might conclude that the superficial and deep layers are both non-differentially involved in the output of a motor signal. Further research is needed to distinguish between the input and output hypotheses by possibly extending the time between the presentation of the probe and motor initiation.

An additional difference between the two studies was the pulse sequence used to measure signals from the layers. We used conventional GE-BOLD, while Finn et al.[29]. used both BOLD and cerebral blood volume (CBV), which resulted in an SS-SI-vascular space occupancy (VASO) contrast[59,60]. The BOLD signal has poorer spatial specificity, but higher sensitivity, as a larger part of the signal comes from larger veins present in the pial surface[61] resulting in a broader point-spread function of the signal in both columns[62] and layers[63] compared to more spatially precise scanning

sequences. VASO has a smaller contrast-to-noise ratio, but localizes activity to both superficial and deep layers, with less bias from large draining veins[59]. The main concern in GE-BOLD are interdependencies in the signal between different layers of the cortex, as a result of the blood draining from deep layers towards superficial layers[64,65]. The superficial layers might thus contain signals from the deep layers as well. One strategy of mitigating the effects of draining veins is to compare differences of tasks, as in the case of our study (high load compared to low load and response compared to abstain trials) and as done in previous GE-BOLD studies; a subtraction of conditions would theoretically remove a linear draining effect[23,24,26,28,66]. Despite employing this method, both superficial and deep layers activated to the motor manipulation. Thus, the linear detrend might have not completely removed the draining vein signal, thus future work can employ more sophisticated models to account for the superficial layer blurring[63,67]; however, preferential deep layer activation has been previously observed in GE-BOLD studies that have not employed these models[23,24,26,68,69].

In summary, we show that superficial layers of the dlPFC are not only crucial for WM maintenance, but in fact are involved in various WM subprocesses by dynamically adapting to current task demands. In addition, we highlight that deep layers seem to have a more complex role in WM than previously understood. Our findings offer new insights into the neural mechanisms of WM and try to bridge the gap between the human and non-human primate WM literature.

## Methods

### Participants

We scanned nine participants (ages 23-36 years, three female) that were recruited from the Max Planck Institute for Human Cognitive and Brain Sciences (MPI CBS) subject database. The sample size was comparable to previous laminar fMRI studies[25,66,70]. All participants gave written informed consent and received monetary compensation for their participation. The study was approved by the ethics committee of the University of Leipzig (441/20-ek). All ethical regulations relevant to human research participants were followed.

### Stimuli and procedure

Following the design of Finn et. al.[29], participants completed four runs of a delayed match-to-sample task (Fig. 1a). In two runs we manipulated the task difficulty by changing the WM load (from now on referred to as the load runs). In the other two runs we manipulated the motor response during the retrieval period (from now on referred to as the motor runs). Each trial began with the presentation of the sample for 3.5 s which consisted of a four-square array where each square was equidistant from the center. A fixation cross was presented centrally. In the load runs either one (low load) or four (high load) items were presented within the squares as the to-be-remembered sample. The items consisted of faces (subcategory: male and female) or scenes (subcategory: indoor and outdoor). The face and scene stimuli were retrieved from the Face-Place database[71]. Faces had neutral emotional expressions. The scenes included indoor and outdoor pictures of houses. All stimuli were converted to grayscale. For each trial, the category and subcategory of the items within the array were the same (e.g., only female faces). In the case of a low load condition, the remaining three squares consisted of Fourier-scrambled items from the same category, in order to preserve the low-level visual features of the overall sample array. Visual masks consisting of Fourier-scrambled images were presented for 200 ms at the location of the four squares after sample presentation (not shown in Fig. 1a). All scrambled images were from the same category as the sample. Motor runs always had four items (i.e., high load) as the sample.

A WM delay period of 13 s followed during which the participants had to remember the sample. At the end of the delay period a probe image (same size as one of the sample squares) appeared centrally with a question mark placed above for 1.2 s. The probe was always from the same category and subcategory as the sample. Participants had 4.2 s to respond using their index and middle finger of their right hand to indicate whether the presented probe was part of the remembered sample. During motor runs

participants had to maintain high-load stimuli for each trial (i.e., high-load WM maintenance, no load manipulation) while we manipulated whether participants had to make a motor response (response trial) or had to abstain from their motor response (abstain trials) during the retrieval period. Identical to the load runs, during the response trials in the motor runs participants had to indicate using their index and middle fingers whether the presented probe was part of the remembered sample. During an abstain trial the probe was still presented; however, an 'X' was placed above instead of a question mark cueing the participants to abstain from responding. If a subject responded during this period on an abstain trial, this was marked as a miss-trial. An inter-trial-interval of 10 s followed the retrieval period.

Stimuli (across both sample and probe) did not repeat within a run. The types of trials within a run were counterbalanced. Categories were counterbalanced for high and low load for each load run. The position of the single item was counterbalanced across the four positions in the low load trials for each load run. The sequence of trials was pseudorandomized for each run; a maximum of three trials of the same type could be subsequently presented. Each trial was 30.7 seconds long. A run had 16 trials and lasted 8 min and 24 s.

Prior to entering the scanner, participants trained by completing eight load trials. To proceed to the scanner participants had to respond correctly to at least 70% of the trials, otherwise, the practice load run was repeated.

### fMRI protocol
MRI data were acquired using a Siemens MAGNETOM Terra 7 T MRI system (Siemens, Erlangen, Germany) using an 8Tx/32Rx head coil (Nova Medical Inc., Wilmington, MA, USA). Anatomical images were acquired using an MP2RAGE sequence (Marques et al., 2010) (TR = 5000 ms, TE = 2.27 ms, 0.75 mm isotropic voxels) at two inversion times (TI of 900 ms, 2700 ms with a flip angle of 3°, 5°, respectively) that were combined to yield a T1-weighted image.

High-resolution functional data were acquired using a T2*-weighted 2D gradient-echo EPI sequence (TR = 2000 ms, TE = 23 ms, 0.8 mm isotropic voxels, flip angle = 80°, 35 slices). Prior to running the high-resolution scans, a whole-brain functional localizer scan using a T2*-weighted 2D gradient-echo EPI sequence (TR = 2000 ms, TE = 18 ms, 2 mm isotropic voxels, flip angle = 75°) was acquired and analyzed to subsequently position the high-resolution partial-brain slab. During this localizer run participants performed an additional load run (see *Stimuli and procedure*). The resulting peak activation defined by the contrast between delay activity and baseline informed the positioning of the high-resolution slab around the expected anatomical location of the dlPFC.

The procedure of the scanning was the following: participants were first scanned using the low-resolution localizer after which the anatomical image was acquired. After semi-manual shimming, participants performed two load runs followed by two motor runs (except for S01 who performed load and motor runs interleaved).

### Preprocessing and co-registration
To obtain accurate reconstructions of gray and white matter surfaces from structural MP2RAGE data, a Freesurfer-based pipeline (Chaimow et al. in preparation,[72]) was applied. First, a T1-weighted image that is spatially homogeneous yet free of extracerebral noise was generated by bias-correcting the INV2 MP2RAGE image before multiplying with the UNI image (MPRAGEize)[73]. Next, a high-quality brainmask was computed by passing this T1-weighted image to CAT12[74] for segmentation and merging the resulting gray and white matter components. Finally, the T1-weighted image was processed using Freesurfer's recon-all pipeline (version 7.1) using the high-resolution flag and substituting Freesurfer's auto-generated brain mask with the CAT12-derived brain mask.

For subsequent ROI selection (see *Definition of ROI*), the segmented structural surfaces of each participant were brought into fsLR164k CIFTI space using ciftify_recon_all, which is part of the Python-based *ciftify* package[75] with expert settings to register to the high-resolution 0.5 mm FSL MNI152_T1 template. Ciftify adapts the post-Freesurfer portion of the Human Connectome Project's (HCP) minimal preprocessing pipeline to non-HCP acquired data[76] and includes surface based alignment (MSMSulc)[77] to HCPs fs_LR space.

Registration between the Freesurfer processed T1-weighted image and the mean image of the third functional GE-BOLD run was calculated using ANTs[78]. The structural image was initially registered using a linear affine transform and subsequently a symmetric normalization (SyN) algorithm generated warp field for distortion correction[79].

Functional scans were motion corrected using AFNI's 3dVolreg using Fourier interpolation to the volume which had the minimum outlier fraction of voxels as calculated by 3dToutcount across all runs (as per standard AFNI preprocessing)[80].

We calculated three equidistant depth bins across the anatomical cortical ribbon in functional space using LAYNII[81]. For smoother layer estimation, prior to layer generation, the anatomical ribbon was upsampled by a factor of five. The upsampled ribbon was then used to generate three equidistant layers. Finally, the layers were downsampled back to the functional images' 0.8 mm-isotropic resolution. Even though we were interested in only two layers (superficial and deep), the middle layer was estimated as a 'buffer' between the deepest and most superficial depths, thus minimizing partial volume effects by excluding voxels where there might have been a large overlap between superficial and deep layers. We applied the affine transformation matrix and warp field generated by ANTs to register the layers to functional space.

### Definition of ROI
We generated two regions of interest using the Human Connectome Project Multi-Modal Parcellation version 1.0 (HCP MMP 1.0) atlas[76]. This surface-based cortical parcellation has been generated using high-resolution, multi-modal data from 210 participants. Surface parcels were transformed from fsLR space to single subject functional space following a combined transformation, estimated by structural-functional registration and MSMSulc[77] surface-based alignment to atlas space (fsLR) in each individual subject. This subject-specific surface-based approach has been shown to substantially improve cortical area localization compared to traditional volumetric approaches and is also superior to other surface-based approaches[82]. We focused all primary analyses on the left dlPFC, following Finn el al.[29]. We defined the left dlPFC by selecting four left-lateralized frontal parcels (8 C, IFJp, IFSa, p9-46v) for which we had sufficient coverage and that correspond to the frontoparietal network[83]. To assess whether our findings were specific to the dlPFC, we selected an anatomically and functionally similar control set of four left-lateralized frontal parcels (FEF, 6r, 46, FOP5) from the cingulo-opercular network (COP)[83]. As an exploratory analysis, we also looked at the dlPFC and the control region in the right frontal cortex, which were defined using the same corresponding parcels in the right hemisphere.

We focused our main analyses on the left dlPFC as this followed Finn et al.[29]. allowing us to have a direct comparison of our results to theirs. Since we decided a priori to focus on the left hemisphere, we also aimed to minimize the distortion of the signal, specifically in that area. The results from the right dlPFC are presented for transparency, as they were part of the acquisition slab and are involved in WM processing.

### Univariate analysis
For each layer, trial time courses were extracted by finite-impulse response modeling using AFNI's 3dDeconvolve with the 'TENTzero' basis function model. A total of 17 regressors were fit to 32 s after the onset of the stimulus. The first and last were zero basis functions, while the middle fifteen free basis functions. We used 'TENTzero' to remove any possible pre-stimulus differences between trial types. In addition, we fit polynomial nuisance regressors up to the fifth order to detrend the signal. One time course was estimated for each trial-type in each type of run (high and low load from the load runs; response and abstain from the motor runs). We used the constant predictor to calculate the percent signal change for each voxel in each trial-type.

https://doi.org/10.1038/s42003-024-06780-8 **Article**

## Multivariate decoding

We performed two decoding analyses: high vs. low load (load decoder) using the load runs, and response vs. abstain (motor decoder) using the motor runs. Decoding analyses were run separately on the superficial and deep layers of the left dlPFC and on the left frontal control regions and corresponding ROIs in the right hemisphere.

We ran the classification across the entire duration of a trial. Time courses of each voxel were high-pass filtered and then temporally z-scored for each run independently. Since the onset of the trial was not TR-locked, we chose the nearest rounded-down TR closest to the onset of the trial and the following 16 TRs (32 s) as a single trial (resulting in a total of 17 TRs). Trial normalization and feature z-scoring was applied to training and testing data independently, thus reducing the classifier's ability to decode based on overall univariate activity differences between classes. Additionally, to increase the signal-to-noise ratio, we averaged the data from two adjacent TRs for each decoding time point. We used a linear support vector machine classifier (LSVMs[84]; libsvm: www.csie.ntu.edu.tw/~cjlin/libsvm/) with a cost parameter of c = 1.

Within a cross-validation fold, we trained on the same number of trials in order to account for the trial transition matrix, which corrected for possible carry-over effects from the previous trial. For example, in the motor decoding, we trained on the same number of retrieval trials where the previous trial was a retrieval or an abstain trial, thus creating four types of trials: retrieval-abstain, retrieval-retrieval, abstain-retrieval, and abstain-abstain. Despite the four types of trials, we ran a binary classification where the current trial determined the label name. The trials chosen for a given fold were randomly selected from all available trials across both runs. The number of cross-validation folds was determined by taking the smallest number of trial types and multiplying them by four. Despite some trials being trained on more than once across folds, each fold had a unique combination of the trials trained and tested on.

## Temporal cross-decoding

To investigate the temporal stability of the multivariate decoding across the delay period, we ran a temporal cross-decoding classification analysis where we decoded the load condition from the superficial and deep layers of the dlPFC. This analysis followed the same classification procedure as the temporal decoding analysis, except each classifier trained on a given time point was tested on data from all time points in turn. This analysis allowed us to see whether the multivariate code at one time point generalizes to other time points and thus to determine whether the underlying code is dynamic or stable. There was no informational overlap between the training and testing data, since the same trials were never included in the two subsets, both when training and testing on the same and different time points.

## Statistics and Reproducibility

Statistical inference for the multivariate analyses was done on empirically generated permuted null distributions. For each decoding analysis, time point, and layer a null distribution was generated by running the same procedure of classification on training data with permuted labels. The analysis was run 250 times for each participant generating a classification accuracy on each iteration. A population null distribution (for each analysis, time point, and layer) was generated by drawing a single sample from each subject's null distribution (resulting in nine samples) and taking the average resulting in a mean $t$-value. This was done 10,000 times. Significance was then determined using a cluster-permutation test[85] where the summed $t$-value of the empirical data clusters was compared to the summed $t$-value of clusters within the generated null distribution. One-tailed tests were run to test for above-chance decoding significance, while two-tailed tests were used for the comparison between layers.

Above-chance decoding of the temporal cross-decoding matrix was tested using a cluster-permutation test[85]. We calculated the summed $t$-value of all above-chance elements within a cluster. Within a matrix, there could have been more than one cluster. A null distribution was generated by a sign permutation test by randomly sampling one of the decoding accuracies

within a given element of the matrix. This was run 10,000 times. The summed $t$-value of the largest above-chance cluster within the permuted data was taken as a statistic for the null distribution. The empirical clusters were then compared to the distribution; all clusters where $p < 0.05$ were deemed to be above-chance.

In addition to calculating an above-chance cluster, we also identified dynamic elements within the cross-decoding matrix. Following previous research[30,42], dynamic elements were defined as off-diagonal elements that had lower decoding accuracies than their two corresponding diagonal elements (e.g., $a_{ij} < a_{ii} \wedge a_{ij} < a_{jj}$). Two cluster-permutation tests were run following the above-chance cluster sign permutation method; in this case all $a_{ij}$ elements were initially subtracted by $a_{ii}$ in one test and $a_{jj}$ in the second test. Only elements that were within clusters from both tests were deemed dynamic. Also, the diagonal entries corresponding to a dynamic element had to both be within an above-chance cluster.

## Data availability
Preprocessed data for all participants and corresponding analysis code are available on the OSF platform (https://osf.io/dtkne/).

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

## Acknowledgements
J.K.D. was supported by the Max Planck Society and BMBF. R.L. was funded by the Wellcome Trust (209139/Z/17/Z). M.A. and J.D. were funded by the UK Medical Research Council (MC_UU_00030/7). J.D.H. was supported by the Deutsche Forschungsgemeinschaft (DFG, Exzellenzcluster Science of Intelligence). N.W. was supported by the Deutsche Forschungsgemeinschaft (DFG, German Research Foundation) – project no. 347592254 (WE 5046/4-2); the Federal Ministry of Education and Research (BMBF) under support code 01ED2210; the European Union's Horizon 2020 research and innovation programme under the grant agreement No 681094. This project also received funding from the Klaus Tschira Stiftung gGmbH foundation (support code GSO/KT18). Stimulus images courtesy of Michael J. Tarr, Center for the Neural Basis of Cognition and Department of Psychology, Carnegie Mellon University, http://www.tarrlab.org/. We thank Rob Mok, Karla Matic, and Laurentius Huber for valuable discussions. The authors acknowledge support by the Open Access Publication Fund of Humboldt-Universität zu Berlin.

## Author contributions
J.K.D., R.L, and D.C. conceived the study; J.K.D. analyzed the data; J.K.D., D.C., D.H., M.A., J.D, and R.L. developed methods; J.K.D. visualized the data; R.L., N.W., and J-D.H, supervised the work; J.K.D. and R.L. drafted the manuscript. All authors reviewed the last version of the manuscript.

## Funding

## Competing interests
The authors declare the following competing interests: The Max Planck Institute for Human Cognitive and Brain Sciences and Wellcome Centre for Human Neuroimaging have institutional research agreements with Siemens Healthcare. N.W. holds a patent on acquisition of MRI data during spoiler gradients (US 10,401,453 B2). N.W. was a speaker at an event organized by Siemens Healthcare and was reimbursed for the travel expenses.
