## [Peer review file · Communications Biology]

Reviewers' comments:

Reviewer #1 (Remarks to the Author):

Degutis et al present a DMTS task in humans, with two separate blocks. In some blocks they compared remembering 4 images vs remembering 1 image (3 masks). In other blocks, they compared respond vs withhold-response conditions. Using layer-specific fMRI and a decoding-style analysis, they concluded that

a) superficial layers have "preferential response to WM load during delay and retrieval".

b) that decoding (after removing the univariate signal) was "dynamic, indicative of separate WM control processes".

c) They found no layer difference in respond-vs-no-response trials.

I am not an fMRI expert. The study seemed to be well executed and analysed. The study is timely and novel as few labs have studied layer-specific WM responses. I had 3 major comments that I think need to be addressed before this manuscript is publishable. I think they are probably all addressable.

1) The authors claim this result supports a "a dynamic account of dlPFC coding" - I am not sure this is warranted, for two reasons:

a) If dlPFC is coding for load, the load may genuinely be different in the different task phases, so it may be the task variable that is dynamic, not the coding. For example, there is obviously no load coding before the stimulus, but this would not be evidence for a "dynamic code". Can the authors show that load is not itself changing?

b) Moreover I don't think they tested the "time x load" interaction which would indeed test for the presence of dynamics. The current claims of dynamics rely on the ability to decode at some times, and the inability to decode at other times. I think the logic may be like drawing conclusions from a null result. For example, if there is more noise, you might not detect a load-code at some timepoints even though it is there. I don't think the authors have statistically tested for a change in decoding over time (unless I missed it?) -- or particularly for an interaction between layer and timepoint.

2) There was a large difference in accuracy between conditions. Could the load effects be due to that? When the authors talk about "decoding load", couldn't this be decoding reaction time? or uncertainty? or response conflict? Also, as far as I can see the "load" is not just a WM load -- there is a difference in attentional load as participants have to simultaneously process 4 items, and presumably there are more eye movements in this condition. Due caution should be exercised in the claims of what the design can claim about "load".

3) The study tries to decode WM load, which is a rather strange thing to do - they provide no mechanistic reason for why the brain should code load in this way. For me it was very counterintuitive to think of an "activity pattern" for high vs low load, after removing univariate effects. I think the manuscript needs to guide the reader through the logic of why one might expect this kind of neural response, what kinds of model might predict it, and what one can conclude about how memory is controlled.

Minor points

4) Figure 1a - should the "+mask" be with the low-load stimulus?

5) Fig 1B - should a 'zero-line' be present? It would be useful also to see (post-hoc) tests of whether there's a significant load effect for *each* of the 6 violins / bunches of dots. (I'm assuming it would be yes,yes,yes,yes,no,no, from the graph?)

6) The methods were unclear whether, in the motor blocks, participants still needed to remember the items themselves. Did they still make a two finger response for present-absent? The authors present only accuracy of responding-vs-not-responding, not memory accuracy, in that condition.

7) Abstract: "indicative of separate WM control processes" - what are the processes? Is this conclusion really warranted? If so what are the arguments / alternatives?

Reviewer #2 (Remarks to the Author):

The study of Degutis et al. investigates the neural mechanisms of working memory in the human dlPFC using 'laminar' fMRI. The study is motivated as a follow-up on the previous findings by Finn et al., who showed the effect of manipulating the working memory content in the superficial layer of DLPFC, and the effect of motor response in the deep layers. The current study uses a very similar experimental design, but focuses on the effect of working memory load. The authors replicate the effect of Finn et al. in the superficial layer for load, but do not find a response effect in the deep layer. They then go on with multivariate analysis and find above chance decoding of load at different time points and in different areas (not only dlPFC, but also control regions). They report higher decoding accuracy in the superficial layer of the dlPFC, and notably also in the right control regions, but never a higher accuracy in the deep layers. The study thus partly replicates the effect reported by Finn et al. However, without a double-dissociation (higher activity/accuracy in the superficial in one condition, and the opposite in another condition), the interpretation of the results as true laminar effects is hampered by the known GE-EPI bias towards the superficial layers.

The study was performed thoroughly. Many methodological details regarding fMRI acquisition and analysis were taken care of. I am, however, not convinced by the presented results of the multivariate analysis. Without them, the study has limited novelty, being quite similar to the study by Finn et al. (2019), but with a lower sample size. The latter could explain some of the negative findings. If the study had sufficient statistical power with a non-replication of the deep layer effect, it would have been more useful to the field of working memory and laminar fMRI. But, as it is, there are only limited conclusions that can be drawn from the results.

Below I make specific suggestions on how the manuscript can be improved.

Major

- I am puzzled by the chosen combination of the research question and analysis approach. The load effect is expected to show as a change in the overall *activity*, rather than as a change the distributed voxel pattern. Therefore, a multivariate analysis added by the authors does not seem appropriate. (Or is there previous research that hints to the multivariate effects of WM load?) On the other hand, the face and scene stimuli that are never mixed within a trial seems to be ideally suited for decoding WM *content* with a multivariate approach. So why not do a univariate analysis for the WM load and a multivariate analysis for the WM content? WM content representations in the DLPFC have not been looked at as far as I know, so this could increase the novelty of the study.

- Related to the above, I currently don't see a lot of meaning in the multivariate results overall. There is an above-chance decoding, and sometimes even laminar differences in decoding accuracy not only in the left DLPFC, but also in the control regions. In my view the MVPA results in the control regions show that whatever is observed in the left DLPFC is not very specific and does not provide any insights into how the DLPFC functions.

- When motivating their sample size of N=9, the authors refer to the previous studies, but strangely, the study of Finn et al., which would be the most relevant here and which has a sample size of N=15, is not among them. The non-replication of Finn et al. effect in the deep layers may be due to lower statistical power of the current study, which the authors should acknowledge or rule out by acquiring more data. Notably, also fMRI effect size in terms of BOLD signal change seems to be lower here compared to Finn et al., which suggests that even larger number of subjects would be needed to show a deep layer effect.

Minor

- Why was the right dLPFC considered a control region? The study of Finn et al. states that they have chosen the left hemisphere due to the verbal nature of the stimuli. The current study uses nonverbal stimuli, so what evidence is there that there will be a difference between left and right dLPFC?

- Can the authors rule out that the difference in decoding accuracy across layers is related to the difference in the number of voxels?

- Methods, fMRI protocol, last paragraph: "participants performed two load runs followed by two motor runs" - why was the order not counterbalanced? Can this explain the null results in the deep layers?

- Figure 1b: there is a typo, it should be "left DLPFC"

- Figure 1a, depiction of single trials. To make it clear to the reader that it was not a 2x2 factorial design, I suggest to present two timelines, one for load runs (on the left) and one for motor runs (on the right), with each option presented within the timeline:

- Load runs: high load/low load stimulus > delay cross > response probe > response cross

- Motor runs: high load stimulus > delay cross > response/abstain probe > "response" cross

- Figure 2 a and c: To make the result presentation more intuitive, I suggest to distribute the data for superficial and deep layers vertically (i.e. superficial high/low load at the top, deep high/low load at the bottom). This will make the comparison of effects across layers easier for the reader.

- The color coding in Figures 2 and 3 is confusing, because in Figure 2 orange and brown represent load conditions, and in Figure 3 superficial and deep layers. Maybe there is a way to stick to the color code introduced in Figure 2 for Figure 3 as well. For example, you can use green and red to show decoding accuracy in superficial and deep layers, and use brown background or plot title color to indicate that the accuracy comes from the load runs. Alternatively, you can keep orange and brown to indicate superficial and deep layers, and plot the univariate load effect (i.e. high-low load difference) in Figure 2, to parallel decoding effect in figure 3 (and do the same for the motor runs).

- Figure 3d: move the y-axis label closer to the plots and away from the 3c plots.

- Supplementary figure 1: the right panel has incorrect tick labels at the x axis, since the depiction is accuracy proportion of **respond** and **abstain** (both for high load)

motor responses. Please also describe what error bars depict and that dots represent mean accuracy of individual subjects.

- Supplementary figure 2: some t-tests are missing degrees of freedom in the brackets; color the titles of plots in a and c, so that it is clearer what layer each plot represents (gray: superficial and red: deep) and that it's consistent with the equivalent main figure.

- It would be nice to include e.g. a vertical rectangle in the plots in Figure 3d to indicate the time point the classifier was trained on (in addition to the plot titles, which are already there).

- Missing citations in the methods section for MSM, FreeSurfer, LIBSVM

- Methods, Univariate analysis: maybe it is my ignorance, but what does the sentence "Fifteen free basis functions and two zero basis functions in the beginning and end were fit to a trial length of 31 seconds" mean exactly? I understand that trial time courses were modelled with FIR. How many timebins/regressors were there in a 31s trial and what was the bin width?

- Overall the manuscript can be improved in terms of readability and grammar. There are multiple long and complex sentences and strange grammatical constructs that are tough to follow. Some examples with suggestions for improvement are listed below:

- Introduction: "In a parallel line of work, instead of elucidating WM contents, studies examined neural activity related to WM load 8–11 ... " – please split this in at least two sentences.

- Introduction, paragraph 2 and throughout the rest of the text: "layers ... preferentially activated to the manipulation" - layers can't *activate to something*, they can *be activated by* something (passive voice)

- Introduction, last paragraph: "retrieval period showed a more significant multivariate load decoding accuracy". Maybe better "we observed higher decoding accuracy during the retrieval period".

- Introduction, paragraph 2: "multivariate population code" - I think just "multivariate code" is more appropriate, because "population code" is typically used to referred to

the information contained in activity of multiple individual neurons (in the context of ephys recordings)

- Introduction, last paragraph, last sentence: at this point, the reader does not know what is meant by dynamic coding, maybe it is better to rephrase it like "Multivariate patterns that distinguished low and high load conditions changed over time".

- Introduction, first paragraph: It would be useful if the authors explained what is meant by "WM manipulation" in the introduction, for readers that are not experts in WM

- Methods, Preprocessing and coregistration, last paragraph: "We applied the transformation affine matrix", should be "We applied the affine transformation matrix "

- Results, Dynamic coding, paragraph 1: "Both univariate and multivariate analyses indicated that superficial and deep layers differentially activate in response to high and low WM load conditions" - "differentially activate" suggests a double-dissociation, where low deep layer shows an effect, but the superficial does not. The study demonstrates effects either in superficial layers only, or in superficial and deep. Therefore, this needs to be rephrased.

- Due to citation formatting, the text sometimes reads odd. E.g., top of page 16: "We focused all primary analyses on the left dlPFC, following [26]". Perhaps sentences which incorporate citations can be rephrased slightly.

- It may be better to call the last trial phase consistently either "response" or "retrieval" phase across block types (in e.g. Figure 2). This would help the reader to understand that you are referring to the same trial phase in different experimental blocks. (I don't insist on this point though)

- In the "Superficial layers preferentially code for WM load in retrieval period" subsection, "timepoints" were mentioned 3 times instead of "time points". Please change it for consistency.

- For readability, the p values can be rounded to 2 or 3 decimal places and with $p < 0.001$ if lower than 0.001

- Effect sizes for statistical tests should be reported

- In the “Both superficial and deep layers non-preferentially code for motor response in retrieval period” section, there is a wrong abbreviation of dlFPC instead of dlPFC

Point-to-point response to reviewers' comments:

We thank the reviewers for their careful assessment of our work that has clearly improved the quality of our manuscript. We have revised the manuscript accordingly which resulted in extensive revisions of the entire manuscript including the Abstract, Methods, Results and Discussion sections as well as additional analyses and revised figures.

We address the reviewer's comments point-by-point below; changes are also highlighted in blue in the revised manuscript.

Reviewer 1's comments:

Reviewer 1 - 1) The authors claim this result supports a "a dynamic account of dIPFC coding" - I am not sure this is warranted, for two reasons:

a) If dIPFC is coding for load, the load may genuinely be different in the different task phases, so it may be the task variable that is dynamic, not the coding. For example, there is obviously no load coding before the stimulus, but this would not be evidence for a "dynamic code". Can the authors show that load is not itself changing?

We thank the reviewer for highlighting that our use of "dynamic coding" was not clear in the previous manuscript.

The term "dynamic coding" is more commonly used to refer to the change of the coding format of WM content (stimulus-specific activity) across time. However, one of the earliest studies looking at dynamic coding decoded different trial types (Stokes et al., 2013). In this study, they observed a context-dependent tuning of the PFC as per the current trial demands. Additionally, the term has also been used to indicate the diverse multivariate patterns that encode different WM control processes within a specific region (Soreq et al., 2019).

In our manuscript, we refer to the same, more broader definition of dynamic coding. As the reviewer correctly points out, we think that it is possible that load varies across different phases of the trial. Hence, in our study, WM load is used as a proxy to investigate the involvement of the dIPFC in different WM control processes since the content of the "load" does change for different trial periods.

To clarify our more broader definition of "dynamic coding", we have added several sentences in the introduction to guide the reader (p. 2):

Fourth, we investigate the stability of this code across the entire duration of the trial, aiming to discern whether well-established dynamic coding properties of the PFC³⁰⁻³⁴ localize to specific layers.

While dynamic coding more commonly refers to changes of the coding format of WM content^{4,30,31,34}, a broader definition of the concept has been used in the past to investigate dynamic changes in multivariate activity underlying different trial types³¹ or to characterize different WM control processes at different trial stages¹⁵. Here, we follow this broader notion of dynamic coding and employ multivariate decoding of WM load to investigate WM control processes across three periods of the WM task (encoding, delay, retrieval) with the aim to understand whether the multivariate patterns underlying these control processes change in a layer-specific manner.

Reviewer 1 - 1b) Moreover I don't think they tested the "time x load" interaction which would indeed test for the presence of dynamics. The current claims of dynamics rely on the ability to decode at some times, and the inability to decode at other times. I think the logic may be like drawing conclusions from a null result. For example, if there is more noise, you might not detect a load-code at some timepoints even though it is there. I don't think the authors have statistically tested for a change in decoding over time (unless I missed it?) -- or particularly for an interaction between layer and timepoint.

We thank the reviewer for emphasizing the robustness of the statistical analysis for dynamic coding.

In our manuscript, we employed one of the most commonly applied techniques to test for dynamic coding in the literature: temporal cross-decoding (Li & Curtis, 2023; Spaak et al., 2017; Stokes et al., 2013; Wolff et al., 2020). The logic underlying temporal cross-decoding analyses for addressing dynamic coding is to look at the generalization of the multivariate code between different time points. For that we decouple the train and test time points where we iteratively train and test on different trial periods. This provides us with two distinct types of elements within the cross-decoding matrix: the diagonal, which indicates the amount of information coding for load within a given time point, and the off-diagonal, which provides the generalization between t and $t+n$. We then run two cluster-permutation tests which allow us to find off-diagonal elements where generalization fails. In each test, we subtracted one or the other corresponding diagonal elements from the off-diagonal elements ($a_{ij} - a_{ii}$ and $a_{ij} - a_{jj}$). An off-diagonal element was deemed dynamic, if both tests were significant ($p < 0.05$) and it was part of the above-chance decoding cluster. Please note that we only take the conjunction of elements present in both clusters, making this test rather conservative. All dynamic clusters found in the left dlPFC have corresponding diagonal elements that have above-chance decoding accuracies (meaning there is information present coding for load). Thus, we do not base our conclusions of dynamic coding on null results; instead, we base it on the lack of cross-generalization of the decoder.

To clarify this point to the reader, we have added the following explanations in the Main Text of the manuscript (p. 9-10):

This analysis provided us with two distinct types of elements within the cross-decoding matrix: the diagonal, which indicates the amount of information coding for load within a given time point, and the off-diagonal, which provides the generalization between two time points. If the trained classifier generalizes to other time points (i.e. high decoding

accuracies of the off-diagonal elements of the decoding matrix), the analysis reveals that the multivariate pattern sustaining load information (i.e. "coding") is stable across time. However, when the diagonal of the matrix, representing training and testing on the same time points, shows higher decoding accuracies than the off-diagonal elements, then we can conclude from this lack of cross-generalization of the decoder, that the multivariate activity is dynamic and varies across time. We examined the generalization of multivariate patterns in response to load in the superficial and deep layers of the dlPFC.

Dynamic elements were calculated as a conjunction of above-chance elements surviving two cluster permutation tests (see Methods for details).

However, we agree with the reviewer that our current cross-temporal analysis does not test for an interaction effect between layers and timepoints. Thus, we thank the reviewer for providing another way of looking at dynamics, and hence we run a time x layer nonparametric repeated measures ANOVA on the multivariate decoding results, which provides us with an interaction term between the two layers and three time periods (encoding, delay, response). More precisely, we take the multivariate decoding accuracies at three time points each (8-12s, 14-20s, and 22-24s). In the left dlPFC we find a significant interaction effect ($p = 0.034$). Interaction effects in the three other ROIs do not reach significance.

This additional analysis has now been added to the Main text (p. 7):

Moreover, we found a significant interaction between the three task periods of encoding (8-12s), delay (14-20s), and retrieval (22-24s) and superficial and deep layers of the left dlPFC ($p = 0.034$, $\eta^2 = 0.35$, non-parametric repeated-measures ANOVA).

[...] (Fig. 3b), but no differences between layers nor an interaction between the three trial periods and layers ($p = 0.70$, $\eta^2 = 0.04$, non-parametric repeated-measures ANOVA). We also decoded WM load from the encoding and early delay period in the superficial layers and deep layers of the right dlPFC and found a difference between layers with higher superficial layer decoding, but no interaction between the three trials periods ($p = 0.97$, $\eta^2 = 0.005$, non-parametric repeated-measures ANOVA) (Fig. S2e). We also decoded load from the right control regions during the encoding and early delay period, but found no interaction between the three trial periods and layers ($p = 0.083$, $\eta^2 = 0.27$, non-parametric repeated-measures ANOVA) (Fig. S2f).

Reviewer 1 - 2) There was a large difference in accuracy between conditions. Could the load effects be due to that? When the authors talk about "decoding load", couldn't this be decoding reaction time? or uncertainty? or response conflict? Also, as far as I can see the "load" is not just a WM load -- there is a difference in attentional load as participants have to simultaneously process 4 items, and presumably there are more eye movements in this condition. Due caution should be exercised in the claims of what the design can claim about "load".

We thank the reviewer for these important comments.

We have now conducted supporting multivariate decoding analyses to address whether differences in accuracy or reaction times might drive the decoding of load in the left dIPFC.

First, we addressed the possibility that our decoding accuracy difference between high and low load trials was due to the difference in accuracy. Here, we removed all trials where the response was incorrect and ran the multivariate decoding on the remaining correct trials. We see a similar pattern of results with superficial layers being more active in the encoding and response periods.

In addition, we ran an analysis to address whether reaction time could play a role in the decoding of load trials in superficial and deep layers. To do so, we calculated the median reaction time of high and low load trials separately, and then labeled the above-median high and low load trials as one class and the below-median high and low load trials as the other class. Trials where no response was made were excluded. We find no difference between the layers. Specifically, we could expect a reaction time difference to drive differential decoding in the response period; however, no such pattern is seen.

Supplementary figure 3: Layer-specific multivariate decoding in the left dIPFC accounting for accuracy and reaction time . a) Multivariate decoding analysis of load when accounted for accuracy. We removed all trials where the response was incorrect and ran the multivariate decoding on the remaining correct trials. We find two above-chance clusters in the superficial layers during the encoding ($p = 0.0003$, $d_{cluster \bar{x}} = 1.18$, one tailed permutation test) and retrieval periods ($p = 0.0051$, $d_{cluster \bar{x}} = 1.15$, one tailed permutation test); one cluster in the deep layers ($p = 0.002$, $d_{cluster \bar{x}} = 0.92$, one tailed permutation test); and one cluster between superficial and deep ($p = 0.014$, $d_{cluster \bar{x}} = 1.50$, two-tailed permutation test). **b)** Multivariate decoding of high and low reaction times. We calculated the median reaction time of high and low load trials separately, and then labeled the above-median high and low load trials as one class and the below-median high and low load trials as the other class. Trials where no response was made were excluded. We find no significant clusters.

We now have added these supporting analyses to the Supplementary Material and refer to the results in the Main Text (p. 7):

These results were not driven by differences in accuracy (Fig. S3a) and reaction time (Fig. S3b) between low and high load trials.

We now address potential confounding factors in the Discussion section of the manuscript (p. 14-15):

With respect to other potential confounding factors, we showed that layer-specific load decoding was not affected by accuracy or reaction time. Equally, we can reasonably exclude the possibility that uncertainty varied significantly between the two load conditions and could have contributed to a differential decoding of load in the two layers, as participants knew how many items they needed to remember at the beginning of the trial. Response conflict also did not drive decoding as in all load trials participants had the options of pressing one of two buttons. Whilst we did ask participants to fixate their gaze on the fixation cross, we did not record eye-tracking and, therefore, cannot rule out the effect of eye movements on load decoding. However, Bastos et al.¹⁸ examined WM-related activity in the prefrontal cortex with laminar electrodes in monkeys and showed a robust deep layer activation processing saccades in the dorsolateral prefrontal cortex. We cannot rule out the possibility for decoding signals related to attentional load and the difference in processing four vs. one item is probably driving the decoding difference during the encoding period. However, this allowed us to investigate the laminar nature of where this type of load is decoded.

Reviewer 1 - 3) The study tries to decode WM load, which is a rather strange thing to do - they provide no mechanistic reason for why the brain should code load in this way. For me it was very counterintuitive to think of an "activity pattern" for high vs low load, after removing univariate effects. I think the manuscript needs to guide the reader through the logic of why one might expect this kind of neural response, what kinds of model might predict it, and what one can conclude about how memory is controlled.

We thank the reviewer for asking us to clarify the mechanistic rationale behind decoding load.

We agree that univariate analyses have been a more common approach to addressing WM load; however, multivariate techniques have been used both in EEG work (Adam et al., 2020; Thyer et al., 2022) and fMRI (Majerus et al., 2016, 2018; Soreq et al., 2019).

We offer two rationales for this approach: First, we propose that during high-load trials, brain activation may spread beyond initially activated regions, potentially recruiting new neural populations (i.e. voxels), when originally-active voxels are already maxed out. Second, drawing from electrophysiological studies in monkeys as well as human fMRI studies, we know that WM load alters coupling between key brain regions involved in working memory processes, which can result in dynamic changes in neural activity patterns in response to variations in load.

More concretely, in monkeys, it has been shown that WM load changes the strength and directionality of neuronal coupling between the prefrontal cortex, the frontal eye fields, and lateral intraparietal cortex (Pinotsis et al., 2019). In the context of human fMRI work, variations in WM load affected between-network coupling among the frontoparietal, ventral attention, and default mode network as well as within-network coupling within the frontoparietal and ventral attention network (Eryilmaz et al., 2020). Even more relevant, a recent fMRI study demonstrated that the multiple demand cortex (Duncan, 2010), a highly domain-general network of regions, of which the dlPFC is part of, dynamically reconfigures its neural pattern of activity and connectivity dependent on WM demands, indicative of a network-coding perspective of WM (Soreq et al., 2019).

As such, it is expected that variations in WM load induce subtle alterations in neural activity patterns within both the superficial and deep layers of the dlPFC, potentially influenced by dynamic changes in directional connectivity within the frontoparietal system and with other brain regions and network of regions. These fine-grained differences within layers could not be discerned via standard univariate analyses but by using decoding analyses leveraging unsmoothed fMRI at ultrahigh-resolution.

We have now added additional sentences in the introduction providing a more detailed mechanistic explanation of investigating activity patterns rather than univariate difference for studying WM load layer-specifically (p. 2):

More recent studies have shown that multivariate activity patterns of the PFC dynamically change depending on WM demands¹⁵, with these changes potentially being driven by altered coupling between the PFC and other key brain regions that process WM load^{16,17}.

Third, drawing from prior research on the impact WM load on the strength and directionality of coupling between the dlPFC and other brain areas^{16,17}, we explore whether WM load induces layer-specific neural activity pattern alterations by investigating the multivariate code underlying WM load in superficial and deep layers.

We also now guide the reader during the results section (p. 7):

Previous studies have used multivariate decoding to investigate control processes of WM memory using load^{15,31,39-42}. Here we followed these studies and trained a binary classifier [...]

We also added a few sentences in the Discussion section to guide the reader about what conclusions can be drawn from layer-specific differences in load decoding (p. 13):

Our findings of WM load-induced layer-specific changes in neural activity patterns in the left dlPFC can be contextualized by previous research in non-human primates and human fMRI studies. More specifically, it has been shown that increased WM load changes the strength and directionality of neuronal coupling between the prefrontal

cortex, the frontal eye fields, and lateral intraparietal cortex ¹⁶, and influences both between-network and within-network coupling among frontoparietal, ventral attention, and default mode networks ¹⁷. This altered network coupling in the left dlPFC may result in subtle variations in neural activity patterns when WM demands change, which has been observed before ¹⁵ and which we localize to the superficial layers of the dlPFC. This observation suggests a network-coding perspective of WM and aligns with the established intra-prefrontal and cortico-cortical connections of the superficial layers in the prefrontal cortex ^{18,33}. In addition, changes in the multivariate patterns of the superficial layers in response to WM load may also be attributed to activation spread beyond initially activated regions, potentially recruiting new neural populations.

Minor points from Reviewer 1:

4) Figure 1a - should the "+mask" be with the low-load stimulus?

The mask refers to the presentation of masks after the presentation of the low-load stimuli as well as high-load stimuli with the aim to mitigate visual aftereffects. The mask does not refer to the scrambled images shown in the low-load stimulus condition only. Figure 1 has now been changed where "+mask (not shown)" has been added to each picture depicting the encoding.

5) Fig 1B - should a 'zero-line' be present? It would be useful also to see (post-hoc) tests of whether there's a significant load effect for *each* of the 6 violins / bunches of dots. (I'm assuming it would be yes,yes,yes,yes,no,no, from the graph?)

We thank the reviewer for that comment and have added a zero-line to the Figure 2B. However, we do not think that running tests against null will yield interpretable results. Following multiple previous laminar fMRI papers that performed a univariate analysis (Finn et al., 2019; Lawrence et al., 2018, 2019) we only tested whether there was a significant difference between layers.

6) The methods were unclear whether, in the motor blocks, participants still needed to remember the items themselves. Did they still make a two finger response for present-absent? The authors present only accuracy of responding-vs-not-responding, not memory accuracy, in that condition.

We thank the reviewer for pointing out this important missing detail. The methods are now updated to state that participants also had to perform the WM task in the motor trials (p. 16):

During motor runs participants had to maintain high-load stimuli for each trial (i.e. high-load WM maintenance, no load manipulation) while we manipulated whether participants had to make a motor response (response trial) or had to abstain from their motor response (abstain trials) during the retrieval period. Identical to the load runs, during the response trials in the motor runs participants had to indicate using their index and middle fingers whether the presented probe was part of the remembered sample.

7) Abstract: "indicative of separate WM control processes" - what are the processes? Is this conclusion really warranted? If so what are the arguments / alternatives?

In the results and discussion we argue that our temporal cross-decoding analysis allows us to investigate the generalization between multivariate load patterns at different time points. The lack of a generalization at different time points is suggestive of different control processes acting on the maintained WM information at different phases of the trial. We argue that the cluster during the encoding period might correspond to the updating of content during encoding; the delay period be the monitoring or contents throughout the delay period or the compression or reorganization of WM contents as a result of increased load demand, while the retrieval period might indicate a comparison process that happens between the novel probe information and the maintained WM information.

We agree with the reviewer that the term "indicative" is too strong, since we do not provide direct evidence of different types of control processes. We have thus changed "indicative" to "suggestive."

Reviewer 2's comments:

Reviewer 2 - The load effect is expected to show as a change in the overall *activity*, rather than as a change the distributed voxel pattern. Therefore, a multivariate analysis added by the authors does not seem appropriate. (Or is there previous research that hints to the multivariate effects of WM load?)

Thank you for this point. We believe that this has been addressed with the response to the last main point of Reviewer 1 (Reviewer 1 - Point 3).

Reviewer 2 - On the other hand, the face and scene stimuli that are never mixed within a trial seems to be ideally suited for decoding WM *content* with a multivariate approach. So why not do a univariate analysis for the WM load and a multivariate analysis for the WM content? WM content representations in the DLPFC have not been looked at as far as I know, so this could increase the novelty of the study.

We agree with the reviewer that an analysis of WM content would provide additional novelty to the study; however, this design is not suited for it for several reasons:

- 1) High and low load trials had a different number of items to be remembered (1 vs. 4 items). Despite the items in high and low load trials being from the same category, the

discrepancy in item quantity poses a challenge for WM content decoding as it has been observed that PFC representations seem to adapt based on the task load at hand. For example, when the task's perceptual demands intensify, the PFC exhibits enhanced neural representations of task-relevant features shown by better decoding accuracy of these features (Woolgar et al., 2011, 2015). This enhancement occurs despite earlier processing areas showing weaker representation of such information (Woolgar et al., 2011). As such, content decoding would significantly interact with load.

- 2) Additionally, restricting our analyses solely to high (or low) load trials poses a considerable challenge for effectively decoding WM content due to an insufficient number of trials per participant. This limitation stems from the significantly lower base rate of content decoding from the prefrontal cortex (PFC) compared to other cortical regions (Bhandari et al., 2018). Specifically, previous studies have indicated that decoding accuracies of WM items in the PFC typically peak around 60%, even with extensive numbers of trials/training data (Bhandari et al., 2018; Christophel et al., 2018; Lee et al., 2013). This unreliable decoding accuracy of WM content may be attributed to several factors, including the absence of a columnar pattern for coding content, as it is encoded in a heterogeneous neural population at a very fine scale (Rigotti et al., 2013). Additionally, the mixed selectivity of conjunctive neurons in the PFC has been hypothesized to contribute to low decoding accuracy of WM content (Manohar et al., 2019). However, higher-level properties such as rule representations (Bode & Haynes, 2009; Reverberi et al., 2012) or task-relevant features (Jackson et al., 2017; Jackson & Woolgar, 2018; Woolgar et al., 2011, 2015) can be decoded from the PFC with high accuracy. Therefore, decoding WM content in our current study would require a larger number of participants and more trials per participant to address these limitations. Hence, we decode WM load as a proxy for control processes across the three phases of the WM task (encoding, delay, retrieval).

These considerations are addressed in another study that we are currently carrying out, in which we use a retrocue paradigm, increase the number of trials per participant, and the sample size itself. This study allows us to investigate content decoding of attended and unattended items held in WM and is pre-registered here: https://osf.io/b5gz9?view_only=b6b75fd8441e4549b0f65cd655420844 (view-only link, as the registration is not yet public). The reason we utilized images of faces and houses in our current study presented in this manuscript is to maintain consistency in stimuli across both studies which facilitates comparability and generalization of our results obtained.

Reviewer 2 - Related to the above, I currently don't see a lot of meaning in the multivariate results overall. There is an above-chance decoding, and sometimes even laminar differences in decoding accuracy not only in the left DLPFC, but also in the control regions. In my view the MVPA results in the control regions show that whatever is

observed in the left DLPFC is not very specific and does not provide any insights into how the DLPFC functions.

We thank the reviewer for asking to clarify the specificity of the multivariate results.

For the left dIPFC, we find that superficial layers show a significant decoding during the encoding and retrieval period and trend towards significance during the delay period. More importantly, we find a significant layer specificity in that we find better load decoding in superficial layers compared to deep layers in the response period as well as trending towards significance in the encoding period. When comparing this to the left control regions, we only find above-chance decoding during the encoding period, but no effects for other trial periods. More importantly, no layer-specific results were identified in the left control regions. To corroborate these observations that descriptively seem highly specific to the left dIPFC, we have now also conducted supporting analyses to investigate any interactions between the three trial periods and the two layers. Here we find a significant interaction effect between the three trial periods and layers in the left dIPFC ($p = 0.034$, non-parametric repeated-measures ANOVA), which was not present in the left control regions ($p = 0.70$, non-parametric repeated-measures ANOVA).

For the sake of completeness and transparency, we have also reported results from the right dIPFC in the Supplementary Material. Please note though, that the right dIPFC is not considered a control region but reported due to its potential layer-specific involvement in the task which we wanted to explore. However in the right dIPFC, we only identify a preferential involvement of the superficial layers compared to the deep layers during the encoding period but not for other trial periods. When comparing the right dIPFC results to the right control regions, we observe also a seemingly preferential involvement of the superficial layers during the encoding period. However, upon closer inspection, we note that the time period of the temporal cluster identified is actually preceding the encoding period and as such seems to be driven by the single negative decoding time point in the deep layers at 4s, which likely negates its significance. Importantly, for both regions we also did not identify any significant interaction effect between trials periods and layers.

To better emphasize the specificity of the multivariate results in the left dIPFC and in line with the reviewer's comment, we now elaborate on our findings by discussing them in conjunction with the results from both the left control regions and the right dIPFC, as well as the right control regions. This detailed comparison provides a more comprehensive understanding of the outcomes in the left dIPFC relative to other regions. Additionally, we have incorporated supporting analyses into the manuscript to strengthen our findings (p. 12-13).

Our results show multivariate laminar activity patterns related to load that are specific to the dIPFC. For the left dIPFC, we find that superficial layers show a significant decoding during the encoding and retrieval period and trend towards significance during the delay period. We find a higher superficial compared to deep layer decoding in the response period as well as trending towards significance in the encoding period. When comparing this to the left control regions, we only find above-chance decoding during the encoding period, but no effects for other trial periods. More importantly, no layer-specific results were identified in the left control region. We also find a significant interaction effect

between the three trial periods and layers in the left dlPFC. The results from the right dlPFC show a preferential involvement of superficial layers during the encoding period. Despite the similar result seen in the right control regions, the laminar difference emerges before the onset of the encoding period and seems to be driven by the single negative time point in deep layers, which likely negates its significance.

Reviewer 2 - When motivating their sample size of N=9, the authors refer to the previous studies, but strangely, the study of Finn et al., which would be the most relevant here and which has a sample size of N=15, is not among them. The non-replication of Finn et al. effect in the deep layers may be due to lower statistical power of the current study, which the authors should acknowledge or rule out by acquiring more data. Notably, also fMRI effect size in terms of BOLD signal change seems to be lower here compared to Finn et al., which suggests that even larger number of subjects would be needed to show a deep layer effect.

We thank the reviewer for pointing out problems due to lower statistical power in our current study.

As the reviewer correctly observes, our fMRI effect size in terms of BOLD signal change seems lower compared to Finn et al. Thus, we agree that a larger number of subjects could theoretically reveal a deep layer effect.

However, there are several reasons why we think that this might not be the case.

- 1) Most importantly, we have recently run a pre-registered study as a direct replication of Finn et al. 2019 (<https://osf.io/txh3s>). In this study, we acquired high-resolution fMRI data from 21 subjects who performed a working memory task identical to Finn et al. (2019). In this direct replication attempt, we were also not able to replicate the deep layer motor effect; in fact we are finding a significant interaction for the opposite effect where the superficial layer also activates significantly stronger to the motor response (see External Figure below and Chaimow et al. *in prep* attached in the Supplementary Material).
- 2) In the context of that replication study, we also ran a bootstrapping analysis to estimate the minimum sample size necessary to achieve a power of at least 80%. For this, we used all single subject datasets from Finn et al. (2019) provided to us by the authors. From that data we generated 1000 datasets of random combinations of single subjects (sampling with replacement) for each tested subject sample size. We then ran the same across layers ANOVA as in Finn et al. (Finn et al., 2019) and counted the proportion of statistically significant interaction effects ($p < 0.05$). This procedure resulted in a minimum sample size of 6 subjects in order to achieve a power of at least 80%. This is also in line with the original pre-print of Finn et al. (2019), where they reported large effect sizes already in six subjects (<https://www.biorxiv.org/content/10.1101/425249v1.full.pdf>). Thus, the effects reported in Finn et al. are so large that even a smaller sample size than ours in the current study should have been sufficient to detect this effect.

Nevertheless, we agree that a more nuanced discussion regarding our sample size is necessary in our current manuscript. To address the reviewer's comment, in the Discussion section, we now acknowledge that based on the results obtained in the current study, it cannot be entirely ruled out that a deep layer effect could be identified with a larger number of subjects. In addition, we have now uploaded the pre-print of our pre-registered replication study on bioRxiv. This provides us with the opportunity to reference this novel data in the context of our Discussion section, offering additional context for interpreting our null results (p. 14):

Equally, our fMRI effect size in terms of percent signal change appears to be considerably lower when compared to Finn et al.²⁹ Whilst they were able to show robust motor effects in a smaller sample size in their original study (N=6)⁷², we acknowledge that our sample is smaller than their final study (N=15). Consequently, we acknowledge that increasing the number of subjects in our study could theoretically reveal a significant motor effect in the deeper layers. However, a recent pre-registered direct replication attempt of Finn et al.²⁹ with 21 subjects found no higher activation of deep compared to superficial layers during the motor manipulation; neither in VASO or GE-BOLD contrasts (Chaimow et al. in prep). Given these mixed results, further data should be acquired to resolve what cognitive operation might drive the deep layer responses in the DIPFC.

Minor points from Reviewer 2:

Why was the right dIPFC considered a control region? The study of Finn et al. states that they have chosen the left hemisphere due to the verbal nature of the stimuli. The current study uses nonverbal stimuli, so what evidence is there that there will be a difference between left and right dIPFC?

We thank the reviewer for pointing out that our manuscript was not clear enough on that point. In our current manuscript, the right dIPFC is not considered as a control region. We consider control regions only as anatomically adjacent parcels localized on the left hemisphere and formally belonging to the cingulo-opercular network.

The focus of our study is on the left dIPFC and this was decided prior to data acquisition. There are a couple of reasons of this:

- 1) In initial lower resolution scans we did not see a high univariate contrast between the task and baseline in the right dIPFC, so we focused on optimizing the acquisition protocol for the left hemisphere.
- 2) At high magnetic fields B0 field inhomogeneities start to play a role affecting data quality across the acquired image and need to be adjusted for using semi-manual shimming. Since we decided *a priori* to focus on the left hemisphere, we also aimed to optimize the signal quality in that area.
- 3) We decided to look into the same regions as in Finn et al. to allow for the comparison of results.

As such, we are only reporting results from the right dIPFC for the sake of transparency and completeness (since it is part of our acquisition slab even though it was not optimized for in the context of shimming). Hence, we placed these results in the Supplementary Material.

As we agree that this might lead to confusion in the manuscript, we have now tried to clarify our reasoning behind presenting results from the right dIPFC in the Results (p. 4):

To allow a systematic comparison with the findings reported by Finn et al.²⁹, all our analyses focused on the left dIPFC (Fig 1b). To assess the specificity of our results, we additionally looked at a set of anatomically adjacent frontal regions in the left hemisphere from the cingulo-opercular network as a control (Fig 1b). We also report results from the right dIPFC in the Supplementary Material due to its involvement in WM processing and as it was covered in our fMRI acquisition slab. As a control for the right dIPFC, we also report results from right-lateralized frontal regions from the cingulo-opercular network. Please note though, that we a priori decided to specifically investigate the left dIPFC and as such optimized our signal in the left hemisphere (see Methods). We found no significant difference in the number of voxels per layer within a given ROI (Fig. S4), allowing comparability of our results across different layers and regions.

As well as the Methods (p. 18):

We focused our main analyses on the left dIPFC as this followed Finn et al. ²⁹ allowing us to have a direct comparison of our results to theirs. Since we decided a priori to focus on the left hemisphere, we also aimed to minimize the distortion of the signal specifically in that area. The results from the right dIPFC are presented for transparency, as they were part of the acquisition slab and are involved in WM processing.

- Can the authors rule out that the difference in decoding accuracy across layers is related to the difference in the number of voxels?

We thank the reviewer for this important comment. We have now run supporting analyses where we statistically compare the size of the layers between our ROIs. Importantly, we do not find a significant difference between the size of the layers in any of the ROIs. This figure is now included as an addition to the supplementary material and referred to in the Results section (p. 4):

We found no significant difference in the number of voxels per layer within a given ROI (Fig. S4), allowing comparability of our results across different layers and regions.

Supplementary figure 4: Number of voxels per layer per ROI. We find no significant difference in the number of voxels between layers in the left dIPFC ($p = 0.152$, $t(8) = 1.582$, $CI^{95} = [-55.7, 299]$), left control region ($p = 0.88$, $t(8) = 0.157$, $CI^{95} = [-211, 241]$), right dIPFC ($p = 0.125$, $t(8) = 1.72$, $CI^{95} = [-52.5, 358]$), right control region ($p = 0.068$, $t(8) = 2.109$, $CI^{95} = [-373, 16.6]$).

- Methods, fMRI protocol, last paragraph: "participants performed two load runs followed by two motor runs" - why was the order not counterbalanced? Can this explain the null results in the deep layers?

The primary reason for not counterbalancing the load and motor runs was the potential of running into scan time limitation of scanning given by the ethical guidelines. When conducting four runs, we consistently approached the maximum limit for participant scanning duration. As such we needed to prepare a priori for cases, where we would have to conduct two scanning sessions per subject. We aimed to ensure that the runs for each analysis (load and motor) originated from the same scanning session. Had we opted for counterbalancing, we might have encountered the scenario where load run 1 occurred in the first session while load run 2 occurred in the second session. Such an arrangement would have introduced complexities during preprocessing, particularly in adhering to the run averaging procedure outlined by Finn et al. (2019). Moreover, it would have posed challenges in selecting the bounding box (field of view) and achieving spatial alignment between runs.

- Figure 1b: there is a typo, it should be "left DLPFC"

- Figure 1a, depiction of single trials. To make it clear to the reader that it was not a 2x2 factorial design, I suggest to present two timelines, one for load runs (on the left) and one for motor runs (on the right), with each option presented within the timeline:

- Load runs: high load/low load stimulus > delay cross > response probe > response cross. Motor runs: high load stimulus > delay cross > response/abstain probe > "response" cross

Figure 1 has now been changed and the corrections and suggestions have been incorporated.

- Figure 2 a and c: To make the result presentation more intuitive, I suggest to distribute the data for superficial and deep layers vertically (i.e. superficial high/low load at the top, deep high/low load at the bottom). This will make the comparison of effects across layers easier for the reader.

We thank the reviewer for the suggestion. It is now implemented for both Figure 2 (seen below) and for Supplementary Figure 2 (right hemisphere).

- The color coding in Figures 2 and 3 is confusing, because in Figure 2 orange and brown represent load conditions, and in Figure 3 superficial and deep layers. Maybe there is a way to stick to the color code introduced in Figure 2 for Figure 3 as well. For example, you can use green and red to show decoding accuracy in superficial and deep layers, and use brown background or plot title color to indicate that the accuracy comes from the load runs. Alternatively, you can keep orange and brown to indicate superficial and deep layers, and plot the univariate load effect (i.e. high-low load difference) in Figure 2, to parallel decoding effect in figure 3 (and do the same for the motor runs).

The colors of superficial layers in all analyses have been changed to gray and the deep layers are now red (as in Figure 2). This has been changed in Figure 3, and all supplementary figures. The titles of the analysis now are colored either brown or purple to reflect load and motor decoding, respectively.

- Figure 3d: move the y-axis label closer to the plots and away from the 3c plots.

An additional "Decoding accuracy" label has been added. One corresponds to the colorbar in 3c) and the other to the y-axis in 3d).

- Supplementary figure 1: the right panel has incorrect tick labels at the x axis, since the depiction is accuracy proportion of **respond**** and ****abstain**** (both for high load) motor responses. Please also describe what error bars depict and that dots represent mean accuracy of individual subjects.**

We thank the reviewer for catching that typo. High/low load has now been changed to respond and abstain. A description about the error bars and dots has been added.

- Supplementary figure 2: some t-tests are missing degrees of freedom in the brackets; color the titles of plots in a and c, so that it is clearer what layer each plot represents (gray: superficial and red: deep) and that it's consistent with the equivalent main figure.

T-test missing degrees of freedom have been added. The titles of the plots have been colored the same way as Figure 3) and also the colors of superficial/deep layers have been changed (as mentioned above).

- It would be nice to include e.g. a vertical rectangle in the plots in Figure 3d to indicate the time point the classifier was trained on (in addition to the plot titles, which are already there).

Gray transparent rectangles have been added to Figure 3d) to indicate the time point at which the classifier was trained. The figure description has been changed accordingly (p. 9):

Transparent gray vertical bars indicate the time points that the classifier was trained on.

- Missing citations in the methods section for MSM, FreeSurfer, LIBSVM

Citations have now been added to MSMSulc, Freesurfer, and LIBSVM.

- Methods, Univariate analysis: maybe it is my ignorance, but what does the sentence "Fifteen free basis functions and two zero basis functions in the beginning and end were fit to a trial length of 31 seconds" mean exactly? I understand that trial time courses were modelled with FIR. How many timebins/regressors were there in a 31s trial and what was the bin width?

We thank the reviewer for this question. We modeled 32 s after the onset of the high/low load stimulus. There were a total of 17 regressors per condition per layer, two of which were set to zero. The logic of having the first and last regressors set to zero for both conditions is to remove any possible pre-stimulus differences, which we do not expect to have before the start of a trial. As per the AFNI documentation (https://afni.nimh.nih.gov/pub/dist/doc/program_help/3dDeconvolve.html) and best practices the bin width should correspond to the TR, which is 2 s in our case. This was calculated as $TR = \frac{\text{trial length}}{\text{bins}-1}$, which gives us 17 bins for 32 s. Since we did not have a TR-locked design (the onset of each trial did not correspond to the onset of the TR), the time in Figure 2 (x-axis) corresponds to a temporal bin of ~1.88 s, which is 32 s divided by 17 bins. Since the methods section was not clear, we have now made the following changes (p. 18):

For each layer, trial time courses were extracted by finite-impulse response modeling using AFNI's 3dDeconvolve with the 'TENTzero' basis function model. A total of 17 regressors were fit to 32 seconds after the onset of the stimulus. The first and last were zero basis functions, while the middle fifteen free basis functions. We used 'TENTzero' in order to remove any possible pre-stimulus differences between trial types. In addition, we fit polynomial nuisance regressors up to the 5th order to detrend the signal. One time course was estimated for each trial-type in each type of run (high and low load from the load runs; response and abstain from the motor runs). We used the constant predictor to calculate the percent signal change for each voxel in each trial-type.

- Overall the manuscript can be improved in terms of readability and grammar. There are multiple long and complex sentences and strange grammatical constructs that are tough to follow. Some examples with suggestions for improvement are listed below:

- Introduction: "In a parallel line of work, instead of elucidating WM contents, studies examined neural activity related to WM load 8–11 ... " – please split this in at least two sentences.

- Introduction, paragraph 2 and throughout the rest of the text: "layers ... preferentially activated to the manipulation" - layers can't *activate to something*, they can *be activated by* something (passive voice)

- Introduction, last paragraph: "retrieval period showed a more significant multivariate load decoding accuracy". Maybe better "we observed higher decoding accuracy during the retrieval period".

- Introduction, paragraph 2: "multivariate population code" - I think just "multivariate code" is more appropriate, because "population code" is typically used to referred to the information contained in activity of multiple individual neurons (in the context of ephys recordings)

- Introduction, last paragraph, last sentence: at this point, the reader does not know what is meant by dynamic coding, maybe it is better to rephrase it like "Multivariate patterns that distinguished low and high load conditions changed over time".

- Introduction, first paragraph: It would be useful if the authors explained what is meant by "WM manipulation" in the introduction, for readers that are not experts in WM

- Methods, Preprocessing and coregistration, last paragraph: "We applied the transformation affine matrix", should be "We applied the affine transformation matrix "

- Results, Dynamic coding, paragraph 1: "Both univariate and multivariate analyses indicated that superficial and deep layers differentially activate in response to high and low WM load conditions" - "differentially activate" suggests a double-dissociation, where low deep layer shows an effect, but the superficial does not. The study demonstrates effects either in superficial layers only, or in superficial and deep. Therefore, this needs to be rephrased.

Thank you. All of these examples have been improved.

- Due to citation formatting, the text sometimes reads odd. E.g., top of page 16: "We focused all primary analyses on the left dIPFC, following [26]". Perhaps sentences which incorporate citations can be rephrased slightly.

Thank you, we have now changed this.

- It may be better to call the last trial phase consistently either "response" or "retrieval" phase across block types (in e.g. Figure 2). This would help the reader to understand that you are referring to the same trial phase in different experimental blocks. (I don't insist on this point though)

All mentions of the "Response" phase of the trial have been changed to "Retrieval."

- In the "Superficial layers preferentially code for WM load in retrieval period" subsection, "timepoints" were mentioned 3 times instead of "time points". Please change it for consistency.

Changed. Thank you!

- For readability, the p values can be rounded to 2 or 3 decimal places and with $p < 0.001$ if lower than 0.001

P-values have now been rounded to the first two non-zero values, and p-values lower than 0.001 are depicted as $p < 0.001$.

- Effect sizes for statistical tests should be reported

Cohen's d have been reported for all t-tests. In the cluster-based analysis, Cohen's d were computed on the average values of the cluster. Partial eta-squared values have been added for the ANOVAs.

- In the “Both superficial and deep layers non-preferentially code for motor response in retrieval period” section, there is a wrong abbreviation of dIFPC instead of dIPFC

Changed. Thank you.

References

Adam, K. C. S., Vogel, E. K., & Awh, E. (2020). Multivariate analysis reveals a generalizable human electrophysiological signature of working memory load. *Psychophysiology*,

- 57(12), e13691. <https://doi.org/10.1111/psyp.13691>
- Bastos, A. M., Loonis, R., Kornblith, S., Lundqvist, M., & Miller, E. K. (2018). Laminar recordings in frontal cortex suggest distinct layers for maintenance and control of working memory. *Proceedings of the National Academy of Sciences*, *115*(5), 1117–1122. <https://doi.org/10.1073/pnas.1710323115>
- Bhandari, A., Gagne, C., & Badre, D. (2018). Just above Chance: Is It Harder to Decode Information from Prefrontal Cortex Hemodynamic Activity Patterns? *Journal of Cognitive Neuroscience*, *30*(10), 1473–1498. https://doi.org/10.1162/jocn_a_01291
- Bode, S., & Haynes, J.-D. (2009). Decoding sequential stages of task preparation in the human brain. *NeuroImage*, *45*(2), 606–613. <https://doi.org/10.1016/j.neuroimage.2008.11.031>
- Christophel, T. B., Jamshchian, P., Yan, C., Allefeld, C., & Haynes, J.-D. (2018). Cortical specialization for attended versus unattended working memory. *Nature Neuroscience*, *21*(4), 494–496. <https://doi.org/10.1038/s41593-018-0094-4>
- Duncan, J. (2010). The multiple-demand (MD) system of the primate brain: Mental programs for intelligent behaviour. *Trends in Cognitive Sciences*, *14*(4), 172–179. <https://doi.org/10.1016/j.tics.2010.01.004>
- Eryilmaz, H., Dowling, K. F., Hughes, D. E., Rodriguez-Thompson, A., Tanner, A., Huntington, C., Coon, W. G., & Roffman, J. L. (2020). Working memory load-dependent changes in cortical network connectivity estimated by machine learning. *NeuroImage*, *217*, 116895. <https://doi.org/10.1016/j.neuroimage.2020.116895>
- Finn, E. S., Huber, L., Jangraw, D. C., Molfese, P. J., & Bandettini, P. A. (2019). Layer-dependent activity in human prefrontal cortex during working memory. *Nature Neuroscience*, *22*(10), 1687–1695. <https://doi.org/10.1038/s41593-019-0487-z>
- Jackson, J. B., & Woolgar, A. (2018). Adaptive coding in the human brain: Distinct object features are encoded by overlapping voxels in frontoparietal cortex. *Cortex*, *108*, 25–34. <https://doi.org/10.1016/j.cortex.2018.07.006>
- Jackson, J., Rich, A. N., Williams, M. A., & Woolgar, A. (2017). Feature-selective Attention in Frontoparietal Cortex: Multivoxel Codes Adjust to Prioritize Task-relevant Information. *Journal of Cognitive Neuroscience*, *29*(2), 310–321. https://doi.org/10.1162/jocn_a_01039
- Lawrence, S. J. D., Norris, D. G., & de Lange, F. P. (2019). Dissociable laminar profiles of concurrent bottom-up and top-down modulation in the human visual cortex. *eLife*, *8*, e44422. <https://doi.org/10.7554/eLife.44422>
- Lawrence, S. J. D., van Mourik, T., Kok, P., Koopmans, P. J., Norris, D. G., & de Lange, F. P. (2018). Laminar Organization of Working Memory Signals in Human Visual Cortex. *Current Biology*, *28*(21), 3435–3440.e4. <https://doi.org/10.1016/j.cub.2018.08.043>
- Lee, S.-H., Kravitz, D. J., & Baker, C. I. (2013). Goal-dependent dissociation of visual and prefrontal cortices during working memory. *Nature Neuroscience*, *16*(8), Article 8. <https://doi.org/10.1038/nn.3452>
- Li, H.-H., & Curtis, C. E. (2023). Neural population dynamics of human working memory. *Current Biology*, *33*(17), 3775–3784.e4. <https://doi.org/10.1016/j.cub.2023.07.067>
- Majerus, S., Cowan, N., Péters, F., Van Calster, L., Phillips, C., & Schrouff, J. (2016). Cross-Modal Decoding of Neural Patterns Associated with Working Memory: Evidence for Attention-Based Accounts of Working Memory. *Cerebral Cortex*, *26*(1), 166–179.

- <https://doi.org/10.1093/cercor/bhu189>
- Majerus, S., Péters, F., Bouffier, M., Cowan, N., & Phillips, C. (2018). The Dorsal Attention Network Reflects Both Encoding Load and Top-down Control during Working Memory. *Journal of Cognitive Neuroscience*, 30(2), 144–159. https://doi.org/10.1162/jocn_a_01195
- Manohar, S. G., Zokaei, N., Fallon, S. J., Vogels, T. P., & Husain, M. (2019). Neural mechanisms of attending to items in working memory. *Neuroscience & Biobehavioral Reviews*, 101, 1–12. <https://doi.org/10.1016/j.neubiorev.2019.03.017>
- Pinotsis, D. A., Buschman, T. J., & Miller, E. K. (2019). Working Memory Load Modulates Neuronal Coupling. *Cerebral Cortex*, 29(4), 1670–1681. <https://doi.org/10.1093/cercor/bhy065>
- Reverberi, C., Görgen, K., & Haynes, J.-D. (2012). Compositionality of Rule Representations in Human Prefrontal Cortex. *Cerebral Cortex*, 22(6), 1237–1246. <https://doi.org/10.1093/cercor/bhr200>
- Rigotti, M., Barak, O., Warden, M. R., Wang, X.-J., Daw, N. D., Miller, E. K., & Fusi, S. (2013). The importance of mixed selectivity in complex cognitive tasks. *Nature*, 497(7451), Article 7451. <https://doi.org/10.1038/nature12160>
- Soreq, E., Leech, R., & Hampshire, A. (2019). Dynamic network coding of working-memory domains and working-memory processes. *Nature Communications*, 10(1), 936. <https://doi.org/10.1038/s41467-019-08840-8>
- Spaak, E., Watanabe, K., Funahashi, S., & Stokes, M. G. (2017). Stable and Dynamic Coding for Working Memory in Primate Prefrontal Cortex. *The Journal of Neuroscience*, 37(27), 6503–6516. <https://doi.org/10.1523/JNEUROSCI.3364-16.2017>
- Stokes, M. G., Kusunoki, M., Sigala, N., Nili, H., Gaffan, D., & Duncan, J. (2013). Dynamic Coding for Cognitive Control in Prefrontal Cortex. *Neuron*, 78(2), 364–375. <https://doi.org/10.1016/j.neuron.2013.01.039>
- Thyer, W., Adam, K. C. S., Diaz, G. K., Velázquez Sánchez, I. N., Vogel, E. K., & Awh, E. (2022). Storage in Visual Working Memory Recruits a Content-Independent Pointer System. *Psychological Science*, 095679762210909. <https://doi.org/10.1177/09567976221090923>
- Wolff, M. J., Jochim, J., Akyürek, E. G., Buschman, T. J., & Stokes, M. G. (2020). Drifting codes within a stable coding scheme for working memory. *PLOS Biology*, 18(3), e3000625. <https://doi.org/10.1371/journal.pbio.3000625>
- Woolgar, A., Hampshire, A., Thompson, R., & Duncan, J. (2011). Adaptive Coding of Task-Relevant Information in Human Frontoparietal Cortex. *The Journal of Neuroscience*, 31(41), 14592–14599. <https://doi.org/10.1523/JNEUROSCI.2616-11.2011>
- Woolgar, A., Williams, M. A., & Rich, A. N. (2015). Attention enhances multi-voxel representation of novel objects in frontal, parietal and visual cortices. *NeuroImage*, 109, 429–437. <https://doi.org/10.1016/j.neuroimage.2014.12.083>

Reviewers' comments:

Reviewer #1 (Remarks to the Author):

- I thank the authors for clarifying the method for establishing lack of cross-decoding, this seems sound to me.

- The authors did a great job of addressing the potential confounders with additional analysis. I am convinced!

- The author's justification of pattern analysis is OK, but:

a) I would just say that the "spread beyond initially activated regions" hypothesis would presumably manifest best as a univariate signal, with added pattern decoding if the initial regions were sufficiently 'rough'/granular.

b) they have still not specified any actual concrete model of WM (cellular, neural network, or representational etc) that would predict a change in pattern without a change in overall activation.

Anyhow these are just matters of interpretation and as such I am happy for the paper to be published.

Reviewer #2 (Remarks to the Author):

The authors comprehensively addressed most of my concerns and provided explanations for design and analysis decisions, which now make more sense. I appreciate the reference to additional unpublished studies, which help to understand the current one.

Only one concern I had raised previously remains unaddressed (-Methods, fMRI protocol, last paragraph: "participants performed two load runs followed by two motor

runs" - why was the order not counterbalanced? Can this explain the null results in the deep layers?)

I understand the authors' need to keep two runs of the same kind together within a limited scanning period, but this could be done by presenting the runs as e.g., load-load-motor-motor in half of the subjects and as motor-motor-load-load in another half. Keeping a fixed order with motor runs at the end is confounded by the order effect, and can potentially explain the absence of expected differences in the motor runs in the deep layers. I think the authors should acknowledge this in the discussion.

Some further (tiny) issues:

Figure 3 C: I guess with the new color-coding, the title of the bottom heat map is supposed to be brown as well (since it is load)!

"Third, drawing from prior research on the impact WM load" - > of WM load

"neither in VASO or GE-BOLD" -> nor in GE-BOLD

Point-to-point response to reviewers' comments:

We thank the reviewers for their subsequent assessment of our work. We hope we have now addressed all remaining comments in our point-by-point response. As in the previous revision, all changes in the manuscript are highlighted in blue.

Reviewer 1's comments:

- The author's justification of pattern analysis is OK, but:

a) I would just say that the "spread beyond initially activated regions" hypothesis would presumably manifest best as a univariate signal, with added pattern decoding if the initial regions were sufficiently 'rough'/granular.

b) they have still not specified any actual concrete model of WM (cellular, neural network, or representational etc) that would predict a change in pattern without a change in overall activation.

Anyhow these are just matters of interpretation and as such I am happy for the paper to be published.

We thank the reviewer for their enthusiastic and positive comments.

- a) We agree that signal spread beyond initially activated regions would manifest as both univariate activation differences and changes in multivariate patterns (depending on the analyses conducted). Given that we performed trial normalization for our decoding analyses (which attempts to mitigate the effect of decoding being driven by univariate activation differences), we think it is plausible that signal spread could be picked up by our multivariate pattern analysis. However, we agree with the reviewer that this should also be seen in the univariate results.

To acknowledge the reviewer's comment, we made the following change in the manuscript:

In addition, changes in the multivariate patterns of the superficial layers in response to WM load may also be attributed to activation spread beyond initially activated regions, potentially recruiting new neural populations, although such signal spread might also manifest in univariate activation differences.

- b) While we do not propose a concrete model of WM that would predict a change of pattern without a change in overall activity in the dIPFC, subtle variations in neural activity patterns when WM demands change have been observed before in both non-human primates (Pinotsis et al., 2019) and humans (Soreq et al., 2019, Eryilmaz et al., 2020, Thyer et al. 2022). When between-network coupling of the frontoparietal, default mode,

and ventral attention networks changes, we would not necessarily expect a univariate signal difference between load conditions but rather an upregulation of certain multivariate patterns underlying between-network connections and a downregulation of others depending on load (Pinotsis et al., 2019). However, we do not propose these mechanisms to be mutually exclusive and we think that both univariate and multivariate analyses can yield interesting insights.

Equally, a model that might both yield a univariate and multivariate change is the recruitment of additional “pointers” in the higher load condition that link the dIPFC to the memorandum that is being stored in sensory areas. Yet, we also agree with the reviewer that *additional* recruitment of pointers may result in increased univariate activity, but such univariate changes might be too subtle to be picked up by a finite impulse response model, while MVPA (support vector machine classification) could be more sensitive as it has implicit voxel selection (in terms of the features/voxels that act as support vectors to maximize the margin of the hyperplane that separates the high and low load classes).

We have now added the pointer hypothesis to the Discussion (p. 13):

Equally, the multivariate pattern in the dIPFC could suggest content-independent pointers⁴⁰ that project to parietal and sensory cortices where the maintained memoranda are thought to be stored^{7,57,58}.

We do acknowledge that our interpretation is mostly speculative and additional research is needed to investigate the nature of the multivariate changes in the code.

Reviewer 2’s comments:

Only one concern I had raised previously remains unaddressed (-Methods, fMRI protocol, last paragraph: "participants performed two load runs followed by two motor runs" - why was the order not counterbalanced? Can this explain the null results in the deep layers?)

I understand the authors’ need to keep two runs of the same kind together within a limited scanning period, but this could be done by presenting the runs as e.g., load-load-motor-motor in half of the subjects and as motor-motor-load-load in another half. Keeping a fixed order with motor runs at the end is confounded by the order effect, and can potentially explain the absence of expected differences in the motor runs in the deep layers. I think the authors should acknowledge this in the discussion.

We thank the reviewer for their comment. There of course might be some way that an order effect explains the lack of differential activity in the layers; however, we think this is highly unlikely due to several reasons. Behaviorally, participants performed at ceiling for the motor runs (Sup. Fig. 1). In addition, we find a main effect of motor response in both the univariate and especially multivariate analyses (Fig. 2 and 3), indicative of the dIPFC’s involvement in that trial

stage. Additionally, Chaimow et al. (in prep) did change the order of the runs and did not see a deep layer effect. Nevertheless, we now acknowledge this possibility in the Discussion (p.15):

Additionally, there may be potential confounds (e.g., increased tiredness, decreased attention) due to our participants performing the motor runs at the end of the scanning session, unlike the alternating run design used in Finn et al.²⁹. However, we believe the effects of differences in run order are negligible to our results, as subjects were performing at ceiling level in the motor runs, and the dlPFC's engagement appears to be robust at that trial stage, as evidenced by both univariate and multivariate analyses.

Some further (tiny) issues:

Figure 3 C: I guess with the new color-coding, the title of the bottom heat map is supposed to be brown as well (since it is load)!

Yes, thank you! It has been changed to brown to reflect the decoding of load in the deep layers of the dlPFC.

“Third, drawing from prior research on the impact WM load” - > of WM load

“neither in VASO or GE-BOLD” -> nor in GE-BOLD

Both have been fixed. Thank you.

References

- Eryilmaz, H., Dowling, K. F., Hughes, D. E., Rodriguez-Thompson, A., Tanner, A., Huntington, C., Coon, W. G., & Roffman, J. L. (2020). Working memory load-dependent changes in cortical network connectivity estimated by machine learning. *NeuroImage*, *217*, 116895. <https://doi.org/10.1016/j.neuroimage.2020.116895>
- Pinotsis, D. A., Buschman, T. J., & Miller, E. K. (2019). Working Memory Load Modulates Neuronal Coupling. *Cerebral Cortex*, *29*(4), 1670–1681. <https://doi.org/10.1093/cercor/bhy065>
- Soreq, E., Leech, R., & Hampshire, A. (2019). Dynamic network coding of working-memory domains and working-memory processes. *Nature Communications*, *10*(1), 936. <https://doi.org/10.1038/s41467-019-08840-8>
- Thyer, W., Adam, K. C. S., Diaz, G. K., Velázquez Sánchez, I. N., Vogel, E. K., & Awh, E. (2022). Storage in Visual Working Memory Recruits a Content-Independent Pointer System. *Psychological Science*, 0956797622109099. <https://doi.org/10.1177/09567976221090923>

REVIEWERS' COMMENTS:

Reviewer #2 (Remarks to the Author):

The authors have addressed all my concerns. I congratulate them on their publication!